# Structure–Activity Relationship of the Dimeric and Oligomeric Forms of a Cytotoxic Biotherapeutic Based on Diphtheria Toxin

**DOI:** 10.3390/biom12081111

**Published:** 2022-08-12

**Authors:** Marcin Mielecki, Marcin Ziemniak, Magdalena Ozga, Radosław Borowski, Jarosław Antosik, Angelika Kaczyńska, Beata Pająk

**Affiliations:** WPD Pharmaceuticals, Żwirki and Wigury 101, 02-089 Warsaw, Poland

**Keywords:** biotherapeutics, cytotoxin, diphtheria toxin, IL-13, inclusion bodies, refolding, disulfide bond, protein oligomerization, MALS, SAXS, LC/MS

## Abstract

Protein aggregation is a well-recognized problem in industrial preparation, including biotherapeutics. These low-energy states constantly compete with a native-like conformation, which is more pronounced in the case of macromolecules of low stability in the solution. A better understanding of the structure and function of such aggregates is generally required for the more rational development of therapeutic proteins, including single-chain fusion cytotoxins to target specific receptors on cancer cells. Here, we identified and purified such particles as side products of the renaturation process of the single-chain fusion cytotoxin, composed of two diphtheria toxin (DT) domains and interleukin 13 (IL-13), and applied various experimental techniques to comprehensively understand their molecular architecture and function. Importantly, we distinguished soluble purified dimeric and fractionated oligomeric particles from aggregates. The oligomers are polydisperse and multimodal, with a distribution favoring lower and even stoichiometries, suggesting they are composed of dimeric building units. Importantly, all these oligomeric particles and the monomer are cystine-dependent as their innate disulfide bonds have structural and functional roles. Their reduction triggers aggregation. Presumably the dimer and lower oligomers represent the metastable state, retaining the native disulfide bond. Although significantly reduced in contrast to the monomer, they preserve some fraction of bioactivity, manifested by their IL-13RA2 receptor affinity and selective cytotoxic potency towards the U-251 glioblastoma cell line. These molecular assemblies probably preserve structural integrity and native-like fold, at least to some extent. As our study demonstrated, the dimeric and oligomeric cytotoxin may be an exciting model protein, introducing a new understanding of its monomeric counterpart’s molecular characteristics.

## 1. Introduction

Immunotoxins (IT) are fusion biotherapeutics composed of two major parts: a receptor-binding moiety and the active toxic payload. The binding part is usually an antibody or a ligand directed towards a specific receptor expressed on the cell membrane, such as IL-13RA2. The active part of the IT is the toxic payload, such as drugs, radioisotopes, toxins, and enzymes [1]. Compared to most drugs, toxins act catalytically, do not develop drug resistance, and can be applied to both dividing and quiescent cells [2]. A major advantage of toxins, when compared to radionucleotides and many small molecule drugs, is a lack of unspecific toxicity to surrounding cells and easier handling. The latter is the main drawback of radiopharmaceuticals [3]. The most commonly used toxins are of plant origin, for example, gelonin and ricin [4], or from bacteria, such as diphtheria toxin (DT) and exotoxin A (PE), produced by *Corynebacterium diphtheriae* and *Pseudomonas aeruginosa*, respectively [5]. Both DT and PE have shown to be highly efficient in killing eukaryotic cells. These molecules can irreversibly modify the mammalian elongation factor 2 (eEF2), thus inhibiting protein translation and inducing cell death [1]. Numerous genetic engineering studies showed that the most potent and non-immunogenic are DT390 and PE38 variants [6]. Hence, DT was selected for the generation of the first immunotoxin, Denileukin Diftitox (Ontak), which was designed to target the IL-2 receptor, which is highly overexpressed in cutaneous T-cell lymphoma (CTCL) [7]. Since then, several toxin-based biotherapeutics directed toward lymphoma and leukemia-specific markers, including CD3, CD25, and CD38, have been introduced. They can be directly injected into the bloodstream, easily reaching the transformed target cells [8,9]. Solid tumors could also be treated with ITs, albeit with lower efficacy, caused mainly by the partial penetration within the tumor mass [10]. Examples of ITs successfully introduced in solid tumor therapy are TE-38, NBI-3001, and Cintredekin besudotox (CB) used in anti-glioblastoma multiforme (GBM) therapy. All of them are fusion proteins containing the PE catalytic domain directed to GBM-specific IL-4R (NBI-3001), transforming growth factor alpha (TGFα) (TE-38), or IL-13RA2 (CB) receptors [11]. In the case of GBM, the blood-brain barrier (BBB) is the major obstacle in systemic drug delivery. Thus, local intra-tumoral immunotoxin administration with the convection-enhanced delivery (CED) method is commonly used [12]. Based on previous results showing the high efficacy of CED-delivered mutated DT390-IL-13 [13], we are currently developing an efficient large-scale production method of the mutated DT390-IL-13 cytotoxin.

Biopharmaceuticals represent one of the most promising frontiers in therapy significant, yet some challenges concerning their production and handling need to be overcome. The living systems producing biotherapeutics are sensitive to minor changes in the production process; hence, even slight differences may significantly affect the biological activity of the therapeutics. Labile biomolecules, including immunotoxins, are prone to degradation when exposed to environmental factors, such as elevated temperature, extreme pH, freezing stresses, organic and inorganic compounds, shear force, and light exposure [14,15]. One of the most common and troublesome effects of external stress is protein aggregation, which can occur at all stages of protein product development, usually leading to reduced bioactivity or increased immunogenicity risk [16,17]. Most pharmaceutically viable proteins must be folded to maintain their activity and molecular crowding, typical for the highly concentrated industrial formulation, is one of the major factors responsible for protein misfolding and aggregation [18,19]. The intrinsic properties of some proteins may also trigger the formation of aggregates. However, factors related to the expression and the purification conditions are believed to play a vital role in this process [20,21]. Reversible aggregation is a significant benefit to product quality in some cases. The native oligomers have higher stability and solubility than their monomeric counterparts, and insulin may be regarded as a canonical example [22]. Nonetheless, in most cases, aggregation, and to some extent oligomerization, is considered a potential detriment to product quality and must be avoided during the manufacture and storage. Although oligomers are common side-products in industrial biotechnology, there are few reports investigating pharmaceutically relevant proteins, and most studies are dedicated to antibodies. For instance, it was reported that monoclonal antibodies (mAb) dimers might form compact or elongated structures, either covalent or non-covalent [23]. Noticeably, some dimeric monoclonal antibodies were reported to be no more immunogenic than unstressed monomers [24].

The oligomerization process of cysteine (Cys)-containing proteins may be promoted or stabilized by forming intermolecular disulfide linkages. In most cases, disulfide bonds stabilize protein structure, and their rearrangement or mis-assembly may destabilize a protein’s structural integrity, leading to the loss of its physiological activity. The dissociation energy of disulfide bond is relatively low and may be further reduced by local peptide flexibility, solvent exposure, and chemical environment facilitating their rearrangements [25]. Despite its reversibility, the formation of non-native cystine linkages often causes abundant oligomerization and aggregation, which has a detrimental effect on the structure and function [26]. Moreover, disulfide shuffling with non-native bonds and free cysteines was also reported for the isolated, stored, or heat-treated bovine serum albumin (BSA) resulting in the formation of dimers and oligomers [27]. It was found that cystine crosslinks largely contribute to forming BSA aggregates, and sulfhydryl interchange reactions probably contributed more to polymerization than sulfur oxidation. Thus, disulfide network rearrangement, also for “structural” linkages, may be more common than expected. Interestingly, the success of attempts to stabilize protein expression and refolding by mutating disulfide-involved cysteines depends on the particular residue and may have both positive and negative effects [28].

Herein we describe the characteristic of the mutated DT390-IL-13 cytotoxin aggregation identified during *E. coli*-based recombinant protein production process and its importance for its biological activity. The exploration of protein-protein interactions background is a fundamental approach to controlling the quality of cytotoxin production and further success of its development as a biopharmaceutic. Using size-exclusion chromatography (SEC) coupled to solution X-ray scattering (SAXS) (SEC-SAXS), we also characterize a similar construct composed of exotoxin A and IL-13 (for brevity, called Exo in the rest of the article) to compare its behavior in the solution as a reference point, since this fusion protein does not incline to oligomerize during purification.

## 2. Materials and Methods

### 2.1. Reagents

Luria-Bertani broth (LB) and MacConkey broth (MB) media, Triton X-100, 5 M NaCl solution, guanidinium hydrochloride, urea, 2,2′,2″,2‴-(Ethane-1,2-diyldinitrilo)tetraacetic acid (EDTA) (0.5 M, pH 8.0), d-(+)-Glucose, protein marker V and peqGOLD were purchased from VWR Chemicals (Gdańsk, Poland). Magnesium sulphate, Isopropyl β-d-1-thiogalactopyranoside (IPTG) and phosphate-buffered saline (PBS), 10× Solution (pH 7.4) were purchased from Alfa Aesar (Kandel, Germany). l(+)-Arginine, l(−)-Glutathione (oxidized), 1,4-Dithioerythritol (DTE) ≥ 99% were purchased from Acros Organics (New Jersey, NJ, USA). Egg white lysozyme, ampicillin sodium salt, Trizma hydrochloride solution (1 M), chromogenic horseradish peroxidase (HRP) substrate, 2,2′-azino-bis(3-ethylbenzothiazoline-6-sulfonic acid (ABTS), 30% hydrogen peroxide solution and citric acid monohydrate were purchased from Merck Life Science (Kenilworth, NJ, USA). RunBlue TEO-Tricine Sodium Dodecyl Sulfate (SDS) Gel 4–12%, RunBlue 20×TEO-Tricine-SDS Running Buffer, RunBlue LDS Sample Buffer 4×, RunBlue 10× Sample Reducer, and InstantBlue Protein Stain were obtained from Expedeon Ltd. (Cambridge, UK). AktaPure 25 FPLC system and prepacked chromatography columns were purchased from Cytiva Life Sciences (Global Life Sciences Solutions, Warsaw, Poland). MTS (CellTiter 96^®^ AQueous One Solution Cell Proliferation Assay) was purchased from Promega (Madison, WI, USA). Cell culture medium, fetal bovine serum (FBS) and penicillin/streptomycin antibiotic mixture were purchased from Biowest (Nuaillé, France). Gentamycin solution was from purchased from TOKU-E (Sint-Denijs-Westrem, Belgium) and albumin from human serum (HSA) was obtained from Merck Life Science (Kenilworth, NJ, USA).

### 2.2. Plasmid and Host Strain

The sequence coding for a single-chain protein consisting of two N-terminal diphtheria toxin domains (DT390) fused to IL-13 was expressed under T7 constitutive promoter in plasmid pWD-MCS in strain BL21(DE3) (New England Biolabs, Ipswich, MA, USA). The expression plasmid was constructed in the Waldemar Dębiński lab (Wake Forest University Health Sciences, Winston-Salem, NC, USA) (details can be found here [29]). Sequence identity of the full-length plasmid was confirmed by MiSeq Illumina sequencing (Genomed SA, Warsaw, Poland). The expression plasmid was propagated in DH5α strain (New England Biolabs, Ipswich, MA, USA). Purity and concentration of the protein preparations were determined spectrophotometrically using DeNovix DS-11 FX, (DeNovix, Wilmington, DE, USA). The recombinant protein consists of 506 amino acids, including initial methionine, with predicted *M*_w_ of 55156 Da, pI = 5.67, and ε = 51715 M^−1^ cm^−1^ at 280 nm, assuming all pairs of cysteine residues form cystines. 

### 2.3. Transgene Expression and Purification of the Recombinant Cytotoxin from Inclusion Bodies

The cytotoxin fusion protein was produced in the T7 expression system with BL21(DE3). Clone selection was performed, and expression kinetics was analyzed to identify the best expresser and optimal post-induction time point (2 h). Bacterial biomass was produced in 5 L shaker flasks by fermentation cultures in the LB Miller Broth media supplemented with 100 mM sodium phosphate buffer pH 7.0, 0.4% glucose, 1 mM MgSO_4_, and 0.2 mg/mL ampicillin. To increase the efficiency of protein production, fed-batch high cell density cultivation was implemented in 5 L Biostat A MO 5 L UniVessel Glass bioreactor (Sartorius Stedim, Kostrzyn, Poland) with initial glucose concentration increased to 1% and 20% glucose as a feeding solution at the induction point in the following conditions, T = 37 °C, pH above 7.0 (adjusted with 0.5 M NaOH), and dissolved oxygen above 25%. Expression was induced at the mid-exponential growth phase with 1 mM IPTG. Inclusion bodies were extracted by suspending the cell pellet in a buffer containing: 0.2 mg lysozyme, 1% Triton X-100, and 0.5 M NaCl. Washed inclusion bodies were solubilized in the buffer: 100 mM Tris-HCl pH 7.4, 7 M guanidinium hydrochloride, 2 mM EDTA. DTE was added to the final concentration of 65 mM. Protein refolding was initialized by a rapid dilution method with disulfide shuffling into the buffer: 100 mM Tris-HCl pH 7.4, 2 mM EDTA, 0.6 M arginine, 0.9 mM oxidized glutathione and continued by dialysis into 20 mM Tris-HCl pH 8.0 with 100 mM urea. The recombinant protein was purified by the two-step chromatography on AKTA Pure 25 M1 FPLC system with Unicorn 7.5.0.1460 software. Dialysate clarified by the high-speed centrifugation was separated by the anion-exchange chromatography in the HiTrap Q HP (5 mL) column with step gradient and 20 mM Tris-HCl pH 8.0—1 M NaCl buffer system. The protein was further purified by the size-exclusion chromatography in the Superdex 200 Increase 10/300 GL column, and stored at −80 °C.

### 2.4. Reducing and Non-Reducing SDS-Polyacrylamide Gel Electrophoresis (PAGE)

Samples denatured in the LDS Sample Buffer (4×) without heating for non-reducing (nrSDS) gels and with a reducing agent and heating for 5 min, at 95 °C, for reducing (rSDS) gels, were loaded into RunBlue precast 4–12% gradient gels. Gels were resolved in TEO-Tricine-SDS running buffer and stained with the InstantBlue protein stain. SDS-PAGE gel scans were analyzed in ChemiDoc MP Imaging Systems with ImageLab 5.2.1 software (Bio-Rad, Warsaw, Poland). The semi-quantitative densitometric analysis assessed the purity of the protein preparations. The apparent *M***_w_** of the selected bands was estimated by the point-to-point (semi-log) regression method. To assess stoichiometry of the resolved oligomeric bands, regression analysis of the measured relative front (*R*_f_) parameter was also performed. The plot of the *R*_f_ normalized to *R*_f_ of the lowest standard band (36 kDa) against the logarithm of standard molecular masses (fitted to the power function of the form *A*x*^B^* where *A* and *B* are model parameters) was used to calculate estimated oligomeric stoichiometries. *R*_f_-estimated stoichiometries were compared to the theoretical stoichiometries by the one-sample *t*-test in the GraphPad Prism 9.3.1 software (LLC, San Diego, CA, USA). A weighted linear least-squares regression of the correlation of the assumed and predicted stoichiometries were also calculated.

### 2.5. Molecular Dynamic Simulations

All protein models were based on two crystallographic structures deposited in the PDB, 1F0L for DT390 and 3LB6 for IL-13. A missing loop in the 1F0L structure between cysteines 187 and 202 was built in YASARA 13.6.16 software. To probe the energy landscape of the fusion protein, five different arbitrary promodels were generated by changing selected dihedral angles in the interdomain linker. The promodels were subjected to molecular dynamics simulation in the NPT ensemble in YASARA (accessed 2 July 2022). Initial cell dimensions were 100/120/100 Å with periodic boundaries. The cell was filled with water with the TIP3P explicit solvent model and neutralized with NaCl. Protonation states for Asp, Glu, His, Lys, and Cys were assigned at pH 7.4 using Ewald summation to compute electrostatic potential [30]. Yamber3 force field [31] with 7.86-A cutoff for non-bonded interactions was applied at T = 298 K. Particle Mesh Ewald algorithm was used for the long-range electrostatic interactions, and pressure was controlled with the water density set to 0.997 g·mL^−1^ with rescaling simulation cell dimensions. DT enzymatic and translocation domains atoms were fixed. The structures were energy-minimized by the steepest descent minimization over 50 ps. The procedure continued by the rescale-atom-velocities temperature control with a Berendsen thermostat and 2 fs time step until at least 20 ns total time was reached. Interdomain energy between DT390 and IL-13 was computed with BindEnregyObj (accessed 2 July 2022) that operates by subtracting the energy of the bound state from the energy of separated parts at infinite distance (the unbound state). The more positive the binding energy, the more favorable the interaction in the context of the chosen force field. The resulting energy, a sum of the potential force field energy components, does not include the solvation effects and ignores entropic terms. Thus, it is only a qualitative estimate to compare the stability of these structures. The hydrodynamic and structural properties were computed in the HullRad 8 software (accessed 2 July 2022) [32], applying full-atom models.

### 2.6. Batch Dynamic Light Scattering (DLS)

Size distributions for the oligomeric fractions were determined by the batch-mode DLS experiments conducted on DynaPro NanoStar 18.5 using auto-attenuation mode (Wyatt Technology Corporation, Santa Barbara, CA, USA). At least 10 isothermal (T = 25 °C) measurements, each with 10 acquisition scans for 5 or 10 s were recorded and analyzed in Dynamics 7.1.9.3 (accessed 2 July 2022) software (Wyatt Technology Corporation, Santa Barbara, CA, USA) using following parameters: PBS refractive index = 1.333, viscosity = 1.019 cP (20 °C), *dn*/*dc* = 0.185 mL·g^−1^. Protein samples in PBS buffer pH 7.4 in the concentration range from 0.8 to 3.3 mg·mL^−1^ were centrifuged before collecting DLS data and were measured in plastic cuvettes with 2 mm optical path length. Dispersity and measurement quality was verified with bovine serum albumin solution in 0.9% NaCl as a monomer control and reference size. Translational diffusion coefficient (*D*_t_), average translational hydrodynamic radius (*R*_h_), and polydispersity parameters were determined using a built-in cumulants algorithm. The polydispersity divided by the estimated *R*_h_ was expressed as the polydispersity percent parameter. A hard-sphere model for globular proteins was applied to estimate the weight-averaged molar mass. Cumulative mass- or intensity-weighted size distributions with resolved particle populations were displayed on the regularization graphs with a non-negative least squares (NNLS) fitting algorithm (DynaLS). 

### 2.7. Size-Exclusion Chromatography Coupled to Multi-Angle Light Scattering and DLS (SEC-MALS-DLS)

The protein samples were filtered and separated at 4 °C on Superose 6 Increase 10/300 GL column in PBS buffer pH 7.4 using NGC Chromatography System (Bio-Rad, Warsaw, Poland) coupled to scattering and quasi-elastic light scattering module (QELS, θ = 90°) detectors MiniDAWN Treos (Wyatt Technology Corporation, Santa Barbara, CA, USA) and refractometer ERC RefractoMax520 (DataApex, Prague, The Czech Republic). QELS temperature probe was enabled using the following parameters: PBS refractive index = 1.331, viscosity = 0.890 cP (25 °C), and *dn*/*dc* = 0.185 mL·g^−1^. All data were collected and analyzed in ChromLab 6.0.0.34 (Bio-Rad, Warsaw, Poland) and ASTRA 7.1.2.5 (Wyatt Technology Corporation, Santa Barbara, CA, USA) software (accessed 2 July 2022). BSA solution in 0.9% NaCl served as a molecular standard for detector normalization, peak alignment and band broadening correction. Debye model with the *R*_Φ_/*K**c = MP_Φ_ formalism and fitting the plot of sin^2^(Φ/2) against *R*_Φ_/*K**c was applied for the data processing of the chromatogram sections. The QELS autocorrelation data were fitted to the single exponential decay. Both *R*_h_ and *M*_w_ were derived for monomeric and dimeric peaks approximately at their full width at half maximum (FWHM) to minimize the variance of non-homogenous fractions. Average *M*_w_, *M*_w_ distributions, polydispersity parameters (*M*_w_/*M*_n_, *M*_z_/*M*_n_), *D*_t_, *R*_h_, and the cumulative and differential weight fractions distributions of the molar mass were also determined.

### 2.8. Size-Exclusion Chromatography Coupled to Small-Angle X-ray Scattering (SEC-SAXS)

All samples were passed continuously through the in-line SEC system (Agilent 1200 HPLC) connected to Superose 6 Increase 3.2/600 column (GE Healthcare, Chicago, IL, USA) and PBS buffer (pH = 7.4) using a flow rate 0.075 mL/min. X-ray scattering measurements were conducted at the B21 beamline of Diamond Light Source (Didcot, UK) using 12.4 keV X-rays at 75 × 75 μm^2^ beam-size with 0.8 × 2 mm photon cross-section and flux 3.0 1012 photon/s, and approximately 620 frames (exposure time = 3 s, *q* range from 0.0031 to 0.38 Å^−1^) were collected per sample using Eiger 4M detector. Initial SEC-SAXS data processing was performed using BioXTAS RAW 2.11 program [33,34]. At least 100 buffer profiles were averaged for background subtraction, and sample profiles were merged for each sample. Single Value Decomposition coupled with Evolving Factor Analysis (SVD-EFA) analyses were performed for each sample after subtraction. All scattering profiles were uncertainty-weighted and stored as column vectors in the data matrix A. The range of each potentially overlapping peak in the data was determined using EFA [35]. Rapid increases in the rank of submatrices A*_i_* are linked to the elution of new components; thus, the start of each peak was identified by plotting of SVs of the initial *n* columns as a function of *n* (which corresponds to the time evolution of an SV during the elution, forward EFA plot). Similarly, a sudden decrease in the ranks of an A*_i_* indicates the end of a given peak, which can be found by plotting SVs of the *n* columns removed from A, also a function of *n* (time-reversal evolution of SVs, backward EFA plot). Pairs of points indicating abrupt changes in the ranks of A*_i_* from forwards and backward EFA plots allow determining concentration windows for a given component. To calculate final *I*(*q*) profiles for each SV, the matrix of basis vectors U is non-orthogonally rotated by matrix R, giving a concentration of each component (c*_i_* column of concentration matrix C), c*_i_* = Ur*_i_*.

### 2.9. SAXS Structural Modeling

*R*_g_, *M*_w_, and *D*_max_ values and *P*(*r*) functions were calculated using BioXTAS RAW directly or via the ATSAS package (version 3.0.5) using RAW as a GUI (accessed 2 July 2022) [34,36]. *P*(*r*) functions were calculated automatically using the BIFT algorithm without modifications [37]. In the Guinier approximation, *R*_g_ was calculated automatically and then tailored to reduce systematic deviation in the distribution of normalized residues, and *R*_g_ based on *P*(*r*) was extracted automatically after computation. Methods of molecular weight calculations: (i) from Porod volume via SAXSMoW 2.0 (accessed 2 July 2022) method, which applies a correction factor to the *V*_p_ based on available data range [38]; (ii) from the volume of correlation as described here [39]; (iii) using machine learning method finding nearest structures in shape and size from PDB database (Shape&Size, implemented in DATCLASS Fortran binary as a part of ATSAS) (accessed 2 July 2022) [40]; (iv) from Bayesian inference with *M*_w_ calculated from methods mentioned above as the evidence to estimate most likely *M*_w_ (implemented in DATMW program as a part of ATSAS) (accessed 2 July 2022) [41]. Normalized Kratky, Porod-Debye, and Kratky-Debye plots and volumetric analysis (calculation of Porod exponent, *V*_p_, and molecular density) were prepared in ScÅtter 4.0 program (accessed 2 July 2022) (Harwell Science and Innovation Campus, Didcot, UK).

### 2.10. SAXS Data Processing

Electron density reconstruction was performed using DENSS program [42] implemented in BioXTAS RAW (accessed 2 July 2022), using previously computed *P*(*r*) as the only input, in slow mode and without any pre-determined symmetry with a number of electrons = 10,000. For averaging and refinement, at least 19 different reconstructions were aligned to obtain each averaged and refined reconstruction. Atomic representations of chimeric proteins were built from existing PDB structures (1DDT, 3LB6 for DT390-IL-13 and 1IKQ, 3LB6 for Exo) in UCSF Chimera (version 1.14) (accessed 2 July 2022) [43]. Missing loops were added using Modeller (accessed 2 July 2022) [44], remaining linkers and additional residues were constructed from Dunbrack2010 rotamers library [45], and such modified structures were merged to create complete atomic models. Local stereochemistry of these models was optimized by molecular dynamics using locPREFMD method [46] implemented in webserver (http://feig.bch.msu.edu/locprefmd/ accessed 2 July 2022) to remove steric clashes and improper rotamers and to add missing hydrogen atoms. Final optimized atomic structures were then used for further studies. Methods of ensemble generations: (i) *BILBOMB*48 implemented in the web server (https://bl1231.als.lbl.gov/bilbomd, accessed 2 July 2022) using 800 conformations per *R*_g_ value, *R*_g_ range ±4 from experimental *R*_g_ determined from *P*(*r*) function, flexible fragments defined manually; (ii) *MultiFoXS*49 implemented in the webserver (https://modbase.compbio.ucsf.edu/multifoxs/ accessed 2 July 2022) using 10,000 conformations with manually defined rigid domains and without imposed distance constraints; (iii) *SREFLEX*50 implemented in *ATSAS online* web server (https://www.embl-hamburg.de/biosaxs/atsas-online/sreflex.php accessed 2 July 2022) in automatic mode using CONCORD (accessed 2 July 2022) during second refinement stage. All structural alignments of conformers and other manipulations or preparation of .pdb files were performed in the ChimeraX program (version 1.2.5) (accessed 2 July 2022) [47]. For each ensemble, a *P*(*r*) function was calculated using BIFT, and these *P*(*r*) functions were compared to appropriate experimental *P*(*r*) functions to find ensembles best fitting to experimental SAXS data. Selected conformers generated by MultiFoXS (accessed 2 July 2022) were subjected to rigid docking in FoXSDock web server (https://modbase.compbio.ucsf.edu/foxsdock/) (accessed 2 July 2022) [48] using default parameters. For the selected dimeric complexes from FoXSDock (accessed 2 July 2022)*,* *I*(*q*) profiles were calculated using the FoXS web server (https://modbase.compbio.ucsf.edu/foxs/) (accessed 2 July 2022) [49], and corresponding *P*(*r*) functions were calculated using BIFT and compared to experimental *P*(*r*) function of the covalent dimer. Monomeric and dimeric atomic models were manually superimposed to ED reconstructions in ChimeraX (accessed 2 July 2022), and volumes of ED maps were adjusted to values estimated from the earlier volumetric analysis. Proposed trimeric and oligomeric structures were manually prepared in ChimeraX (accessed 2 July 2022) using dimers from FoXSDock (accessed 2 July 2022) and superimposed to ED maps, their *I*(*q*) profiles were calculated using FoXS (accessed 2 July 2022) webserver and *P*(*r*) functions were calculated by BIFT and compared to corresponding experimental *P*(*r*) functions. All visualization of protein structures and ED maps were prepared in ChimeraX (accessed 2 July 2022).

### 2.11. Disulfide Bonds Mapping and Structural Analysis

Monomeric and oligomeric fractions were analyzed on a tandem LC-MS/MS system composed of the nanoAcquity UPLC (Waters Corporation, Milford, MA, USA) coupled to the Q Exactive Hybrid Quadrupole-Orbitrap mass spectrometer (Thermo Fisher Scientific, Waltham, MA, USA). Protein samples were non-reductively trypsin-proteolyzed for 10 h, acidified with trifluoroacetic acid (to 0.1%), and resolved by C-18 reverse-phase liquid chromatography on the trapping column (M-Class Symmetry C18 Trap Column, 100 Å, 5 µm, 180 µm × 20 mm) (Waters Corporation, Milford, MA, USA) and subsequently transferred online and resolved on the nano-HPLCRP-18 separation column (M-Class Peptide BEH C18 Column, 130 Å, 1.7 µm, 75 µm × 250 mm) (Waters Corporation, Milford, MA, USA) using a linear gradient of acetonitrile with 0.1% TFA (0–35% per 160 min, 0.25 mL/min flow rate and T = 40 °C). Data collection parameters were: data-dependent acquisition (DDA) and Top12 mode, scan range 300–2000 *m*/*z*, HCD fragmentation, mass resolving power 70,000 for MS1 and 35,000 for MS2, maximum ion accumulation time 60 ms for MS1 and 300 ms for MS2, automatic gain control (AGC) target value 10^6^ for MS1 and 2∙10^5^ for MS2, isolation window 3.0 *m*/*z*, normalized collision energy (NCE) 27, and dynamic exclusion of 30 s. Raw data processing, peptide and crosslink identification, and hit assembly were accomplished in MaxQuant 2.0.3.1 (accessed 2 July 2022) [50] and the Andromeda search engine [51]. Intensity is calculated as the summed up eXtracted Ion Current (XIC) of all isotopic clusters associated with the identified peptide sequence. The quality of recorded peptide spectra matches was ranked according to the binomial-probabilistic Andromeda scoring model and posterior error probability (PEP) with 5% threshold. The target-decoy-based 1% false discovery rate (FDR) cutoff thresholds for a peptide, a modified peptide, and a protein were applied. Prediction of aggregation propensity was computed using Aggrescan3D 2.0 method [52] on the webserver (http://biocomp.chem.uw.edu.pl/A3D2) (accessed 2 July 2022) in dynamic mode and 10 Å as a distance for aggregation analysis. Using default parameters, flexibility analysis was performed using CABS-flex 2.0 method [53] on the webserver (http://biocomp.chem.uw.edu.pl/CABSflex2) (accessed 2 July 2022). The calculated conformational ensemble was manually docked to ED reconstruction in ChimeraX (accessed 2 July 2022). 

### 2.12. Titration Enzyme-Linked Immunosorbent Assay (ELISA)

Titration ELISA with the extracellular domain of IL-13-RA2 human recombinant receptor protein (ACROBiosystems, Newark, DE, USA) was performed at room temperature (RT). Pierce Nickel Coated Plates, Clear, 8-Well Strips (Thermo Fisher Scientific, Waltham, MA, USA), pre-blocked with BSA, were incubated with 100 µL of the 1 µg·mL^−1^ receptor in Dulbecco’s PBS (DPBS) (Biowest, Nuaille, France) for 1 h. Strip wells were subsequently washed with DPBS and blocked with 2% skimmed milk in DPBS overnight, at 4 °C. Different fractions of the cytotoxin ligands, monomer, dimer, and lower oligomers, were added to the wells in serial dilutions in the concentration range 100–0.0001 µg·mL^−1^ for monomer and dimer and 50–0.00005 µg·mL^−1^ for lower oligomers, all in the presence of 0.1% BSA, each in 6 replicates and incubated for 2 h. Next, the strips were washed and incubated with the primary antibody (working dilution 1:165) for 2 h and the secondary (working dilution 1:3000) antibody for 1 h. The primary antibody was rabbit polyclonal anti-IL-13 antibody 0.1 mg/mL (OriGene, Rockville, MD, USA) and secondary—goat anti-rabbit IgG Fc, HRP conjugated, polyclonal (Agrisera AB, Vännäs, Sweden). After thorough washing, the strips were developed with 0.222 mg·mL^−1^ ABTS in 50 mM sodium citrate pH 4.0, supplemented with 0.05% H_2_O_2_. Then, 405 nm absorbance data were collected in the Tecan Infinite M200 PRO plate reader and the Magellan *7.2* software (accessed 2 July 2022) (Tecan, Mannedorf, Switzerland) and fitted to the sigmoidal four-parameter dose-response model in the GraphPad Prism 9.3.1 software (GraphPad Software, San Diego, CA, USA). The following parameters were derived: bottom, top, Hill coefficient, and LogEC_50_ with 95% confidence interval and adjusted R2 for monitoring the weighted (1/Y) non-linear least-square regression quality. The null hypothesis was examined with the extra-sum-of-squares F test. The ordinary one-way ANOVA performed the further multiple pairwise comparisons of the LogEC_50_ parameter with Tukey’s correction and reporting *p*-values. The data sets were also fitted to the quadratic regression model to derive the apparent dissociation constants (see [54] for details).

### 2.13. In Vitro Functional Assay

Human GBM cell lines U-251 MG and LN229 were obtained from Wake Forest Baptist Medical Center Comprehensive Cancer Center and authenticated with STR analysis by ATCC. Cultures of U-251 (overproducing IL-13RA2) or LN229 (very low expression of IL-13RA2) cells were maintained in DMEM HG medium supplemented with 10% (*v*/*v*) fetal bovine serum and antibiotics: penicillin/streptomycin (1% *v*/*v*) and gentamycin (0.06% *v*/*v*). Cells were grown at 37 °C in a humidified atmosphere with 5% CO_2_. Cell viability was determined by the MTS method. Then, 1 × 10^3^ cells were plated per well into a 96-well plate in quadruplicates for each tested concentration and allowed to attach overnight. Next, the medium was replaced with fresh medium supplemented with HSA (final concentration: 0.1% *v*/*v*) and 0.1, 1, or 10 ng/mL of IL-13-based cytotoxins (monomer, dimer, and oligomers) for 72 h. Then, 20 mL of MTS solution was added to each well. After 4 h of incubation, absorbance was measured at 490 nm (with reference wavelength 670 nm) in a TECAN Infinite M200Pro microplate reader (Tecan, Mannedorf, Switzerland). Data were obtained from three independent experiments and presented as a percentage of the control. Data were analyzed using GraphPad Prism 8 software (GraphPad Software, San Diego, CA, USA). Two-way ANOVA followed by Tukey’s multiple comparison test was used to calculate the *p*-value and determine the statistical significance of the difference in the measured variables between the control and tested groups. A difference between control and tested groups was recognized as significant when *p* < 0.05.

## 3. Results and Discussion

### 3.1. Preparation and Purification of the Cytotoxin

The fusion cytotoxin was expressed in *E. coli* without an affinity tag, in the process consisting of three major steps. After a short induction (2 h), the recombinant protein was recovered from inclusion bodies and solubilized in guanidium hydrochloride and reducing environment. The solubilized protein was subjected to refolding by the rapid dilution method and subsequent dialysis. The refolded and soluble fraction was separated and purified by anion-exchange chromatography (IEX) followed by size-exclusion chromatography (SEC). The production yield was 0.6–1.9 mg·L^−1^ at 90–99% purity, which was determined by densitometry. Later, the fermentation was switched from shaker flasks and orbital shaker to the medium size bioreactor unit, increasing yield up to 6.1–6.9 mg·mL^−1^ and achieving high reproducibility. Pressure homogenization was also incorporated into the purification protocol, and the cytotoxin recovery was in the range of 3.3–6.8% for monomeric cytotoxin. However, we found that a noticeable amount of the protein forms oligomers and aggregates during preparation. Since it reduces the yield and purity of the final product, we decided to characterize their structural and functional properties. 

Figure 1 depicts the results of IEX and SEC chromatography. SDS-PAGE resolved the obtained fractions. We found that the main constituent of all fractions is the recombinant cytotoxin, and both IEX and SEC separate different soluble oligomeric states of the same protein. The monomeric cytotoxin was eluted at 15% buffer B during IEX and at SEC elution volume (*V*_SEC_) of 14.4 mL, whereas for the dimer, these parameters were 20% and 12.2 mL, respectively. The oligomers eluted with IEX ionic strength above 20%, usually 25% and 30%, and *V*_SEC_ between 8 and 12 mL (peak maximum at 9.5 mL). The higher the oligomers, the lower the *V*_SEC_ with aggregates eluting closely to the exclusion limit of the column. These higher molecular forms are easily disrupted in reducing conditions into monomers, which indicates that oligomerization is cystine dependent, and oligomers are probably formed during refolding step. 

Assuming that both dimer and oligomers are built by native and physiologically active monomers, we undertook some attempts to recover monomeric protein. Firstly, we tested a straightforward reduction using 50–100 tris(2-carboxyethyl)phosphine (TCEP)-to-monomer molar ratio, which was found to be sufficient to disrupt oligomers. However, the reduction led to excessive aggregation, making this strategy ineffective (Figure 1B). The second strategy was to use oligomers as potential recyclable substrates for the solubilization step of our biomanufacturing process (Figure 1B). The chromatographic and SDS-PAGE gel profiles are similar to those obtained routinely, with a total yield of 6.8% (pure monomer) and some contamination by dimer and oligomers. Expression and purification of Exo protein were performed using the analogical protocol (see Appendix A).

Our purification protocol is relatively simple and costs less than eukaryotic expression systems, and the recombinant protein requires no tag removal. Nevertheless, the production yield is relatively low (~10%), and the protein is prone to oligomerization. Although the oligomers can be resolved by a combination of IEX and SEC, their formation results in monomeric product depletion, lowering the method’s efficiency. Noticeably such behavior is a well-recognized phenomenon in the manufacturing of therapeutic antibodies [55]. However, it has not been reported for fusion cytotoxins yet. 

Although the utilization of many novel expression systems for biomanufacturing biotherapeutics is constantly growing [56], protein expression in *E*. *coli* is still most prevalent in both industry and academia, also being usually the first-choice method to test protein production. Protein expression into inclusion bodies has long been distinguished as facilitating tag-free and protease-resistant protein manufacturing. More than 20 recombinant proteins are manufactured by recovery from inclusion bodies and refolding into soluble active products for industrial and biomedical applications [57]. 

### 3.2. Stoichiometry of the Oligomers

Dimers and low mass oligomers were resolved by non-reducing SDS-PAGE, whereas only a single band representing monomeric form was visible on reducing SDS-PAGE. This straightforward assay confirmed that oligomerization is cystine-dependent and assessed their electrophoretic mobilities (*R*_f_) and stoichiometries (Figure 2).

The first two oligomeric bands were in the range of 250 kDa. Their apparent molecular masses (*M*_W-app_) were estimated by fitting a power function as a function of *R*_f_. Their values were 62.7 ± 3.1, 123.1 ± 6.4, and 243.4 ± 6.1 kDa, indicating the following stoichiometries: monomer, dimer, and tetramer, respectively. To estimate stoichiometries of other oligomers, we applied analysis of their relative fronts (Figure 2), which resulted in the following numbers: 2.1 ± 0.1, 4.1 ± 0.1, 5.9 ± 0.4, 7.7 ± 0.5, and 9.5 ± 0.5. These were compared with hypothetical stoichiometries by the one-sample *t*-test (Figure 2). Although the most statistical significance may be attributed to the envisioned stoichiometries: 2, 4, 6, 8, and 10, more uncertainty was detected for octamer and decamer because of the reduced resolving power of SDS-PAGE. Moreover, dimers and tetramers are prevalent forms of DT390-IL-13 complexes, and the higher oligomers are relatively scarce. The obtained results suggest that the cytotoxic complexes are formed by dimeric units, which was further confirmed by the results of MALS and SAXS.

### 3.3. Computational Structural Studies

Comparative structural studies on our fusion cytotoxin in comparison to the native diphtheria toxin (DT) were analyzed by molecular dynamics (MD) simulations (Figure 3 and Appendix A). Five hypothetical models of DT390-IL-13 and a model of DT based on X-ray structures were subjected to 20 ns simulations. 

Values of time-averaged binding energies between IL-13 or the DT receptor binding domain and the enzymatic/translocation domains were significantly lower for the recombinant cytotoxin than for DT (2489 kcal·moL^−1^), with the highest for Model 4 (1465 kcal·moL^−1^) (Appendix A). Since the DT model was directly based on the low-energy crystallographic structure, it is expected to be more stable than the fusion protein with non-optimal interactions between both domains. The computed volumetric parameters ($\bar{V}$, *R*_0_) are based on the single amino acid properties, so they are identical for all the models. Other parameters, including *R*_g_, *D*_max_, *a*/*b*, *f*/*f*_0_, and *phi*_c_ for both proteins, are similar and confirm their average prolate geometry (see Appendix A for the details). The slight differences probably were caused by replacing the binding domain with IL-13 in the DT390-IL-13 fusion protein. The values of *R*_g_ and *R*_t_ for the monomeric cytotoxin models are in the ranges of 26.2 to 28.1 Å and 33.3 to 34.5 Å, respectively, whereas for DT, their values are 25.4 Å and 32.8 Å, respectively. The highest values of shape parameters were calculated for the “extended” Model 1 and the lowest—for the most energetically stable Model 4. The asphericity coefficient is higher for DT (0.21) than for the cytotoxin models (0.15–0.17). It can be concluded that swapping the receptor-binding domain for the IL-13 domain in the cytotoxin monomer may have a negligible influence on the molecule’s size and shape. Nevertheless, the recombinant cytotoxin is expected to possess a less compact structure than the native toxin, due to the lack of specific intramolecular interaction between both domains. 

### 3.4. Light Scattering Spectroscopy—DLS

Hydrodynamic properties, including *R*_h_, dispersity, and modality, were determined by the batch DLS measurements using BSA as a reference (Figure 4 and Appendix A). BSA sample is characterized by narrow and consistent intensity- and mass-averaged size distributions with a negligible mass fraction (about 0.3%) of particles with *R*_h_ above 10 nm. The *R*_h_ value for BSA was 4.1 ± 0.1 or 3.8 ± 0.1 nm for cumulant and regularization methods, respectively, which indicates a monomodal sample with a low polydispersity (Appendix A). On the contrary, for the monomeric fraction, large particles (>100 nm) were predominant, whereas small ones (<10 nm) contributed only about 1.0%. This sample demonstrated much higher cumulant *R*_h_ (239.9 ± 10.6 nm) and its polydispersity (53.0 ± 4.3%) but similar *R*_h_ values for Peak 1 (4.2 ± 1.3 nm), which corresponds to monomeric particles. Comparable results were obtained for the dimer sample, where the Peak 1 radius was slightly higher (5.1 ± 1.0 nm). Nonetheless, both samples are statistically indistinguishable by DLS. These results suggest a tendency of the cytotoxin molecules to aggregate and oligomerize in the solution. In the case of the oligomer fraction, the particles with radii in the range of 5–50 nm were responsible for 93.0% of the intensity. Mass-averaged distribution for the oligomers revealed that about half the mass of scattering particles come from smaller scatterers with the *R*_h_ = 2.4 ± 0.9 nm and another half from larger scatterers with *R*_h_ = 14.9 ± 1.5 nm suggesting bimodal distribution. On the other hand, the characteristic of Peak 2 is similar to cumulant data (*R*_h_ =12.6 ± 0.4 nm, polydispersity 3.0 ± 0.1 nm), suggesting monomodality with the predominant oligomeric particles (Peak 2). These findings suggest a general tendency for oligomerization and aggregations of DT390-IL-13 with high-mass particles in both the monomer and dimer samples.

### 3.5. Light Scattering Spectroscopy—MALS

To reduce the ambiguity of the results of batch DLS, we applied SEC-MALS-DLS to analyze the size distribution of DT390-IL-13 (Figure 4C,D) from different fractions. The separation of BSA, used as a control, resulted in peaks corresponding to monomer (95.1%), oligomers (4.1%), and aggregates (0.8%). The cytotoxin monomer sample was resolved into three peaks, monomeric (93.6%) and two oligomeric (6.4%). The measured *M*_w_ for the monomer was 52.0 kDa showing high similarity to the actual *M*_w_ (55.1 kDa). The *R*_h_ value determined by MALS (3.3 ± 0.1 nm) was more precise than the DLS results. For the dimeric sample, three MALS peaks were found, corresponding to the following molecules: dimer (88.8%), oligomers (10.7%), and aggregates (0.5%), with *M*_w_ and *R*_h_ for the dimeric peak of 104.9 kDa and 4.7 ± 0.2 nm, respectively. The dimer-to-monomer *M*_w_ ratio was 2.0. Dispersity parameters, *M*_w_/*M*_n_ and *M*_z_/*M*_n_ were virtually the same for the dimeric fraction (1.001) in comparison to the monomeric fraction and BSA (1.000). These parameters were higher for the full peaks of the monomer and dimer than BSA, which suggests slight amounts of oligomers. In the case of the oligomeric sample, four different scatterers sets were found *a* (*M*_w_ 258.5 kDa, *R*_h_, 7.9 + 0.5 nm, *t*_el_ 32 min), *b* (high *M*_w_, *R*_h_ 8.9 + 0.7 nm, *t*_el_ 25–30 min), *c* (high *M*_w_, *R*_h_ 11.8 + 1.2 nm, *t*_el_ 20–25 min) and *d* (*M*_w_ beyond the detection limit), which indicate high polydispersity (Appendix A). Sets *b* and *c* had the highest mass contribution (45.0% and 34.2%, respectively) and their *M*_w_ was more sparsely distributed when compared to *a*. 

Separated oligomeric fractions were subjected to nrSDS-PAGE, which resolved monomer, dimer, and oligomers (Figure 5). As expected, oligomers *M*_w_ was inversely proportional to *t*_el_. The oligomers, as well as aggregates, were easily monomerized after reduction by DTT. Protein quantity is decreasing for the fractions of high oligomers to the identified Peak 3 on SEC-MALS (Appendix A), and the last three fractions are devoid of the protein. This indicates that particles present in this peak are unstable, and experimental manipulations accelerate their denaturation and precipitation. The *M*_w_ values determined by MALS and similar methods are weight-averaged (Equation (1)), and the accurate distribution of different particles in the polydisperse system may be challenging to estimate without additional experimental evidence.
(1)$$\bar{M_{w}}=\frac{\sum c_{i}M_{i}}{\sum c_{i}}$$

SDS-PAGE analysis of the low-oligomeric fractions indicated that the mass fraction is inversely proportional to the oligomer stoichiometry. The oligomer fractions also contain some amount of the monomer and the dimer. This way, the MALS-determined *M*_w_ of 258.5 kDa could be potentially resolved into its ingredients, monomers, and oligomers with the 2*n* stoichiometry. Assuming the monomer and dimer contents are each less than 12–20% and the octamer/decamer are at the low level of 1–2%, the equation-calculated tetramer and hexamer fractions would be in the range of approximately 26–43%. These results are in accordance with DLS and were further confirmed by SAXS. 

Oligomeric samples are polydisperse, and oligomerization generates rather spheroidal than rod-like particles. It is believed that most proteins are prone to (self)-assembly, especially in higher concentration and non-biological environments, and sometimes even a single point mutation may increase such tendencies. Our findings suggest that the oligomerization may not be governed only by a random disulfide rearrangement and misassemble but is also supported by non-specific intermolecular interactions in the vicinity of Cys residues involved in the formation of disulfide bonds. The notion may be supported by the fact that the Exo protein showed no oligomerization tendency (Appendix A), at least in the preliminary experiments, yet still may form aggregates. Thus, oligomerization might be triggered by the presence of solvent exposed Cys residues in the flexible region of DT390.

### 3.6. Small-Angle X-ray Scattering

#### 3.6.1. Decomposition of SAXS-SEC Data

For further biophysical 1analysis, we applied SEC-SAXS to achieve better buffer subtraction, separate different molecular species, and remove aggregates. A set of samples containing different oligomeric states of DT390-IL-13 and Exo proteins was (datasets 1.0–5.0) ubjected to SEC-SAXS. Initial *I*(*q*) profiles indicated flexibility and potential aggregation in each sample subjected to SEC-SAXS (Appendix A). 

Despite the application of SEC-SAXS to separate different oligomeric states of the proteins, initial data analysis, e.g., the shape of scattering profiles and *P*(*r*) functions of datasets 1.0–5.0, indicated polydispersity and aggregation. In order to identify a number of scattering components and their relative contributions, Singular Value Decomposition (SVD) was used. SVD is a data reduction technique based on the factorization of data matrix A to extract its main components. SVD decomposes the scattering data into orthonormal components, known as singular vectors, ranked in order of significance by the singular values (SV). Detailed mathematical descriptions can be found elsewhere [58], and a brief description is available in ESI. Our SVD analysis indicated a mixture of overlapping scattering species in each fraction, rendering straightforward data processing complex since automatic methods were unable to discern sets of scattering profiles suitable for further processing (see Appendix A). To overcome this hurdle, we applied Evolving Factor Analysis (EFA), a model-free data deconvolution method suitable for time-dependent datasets [35,59]. Using EFA, we resolved overlapping signals from all SEC-SAXS datasets, except 2.0, when it led to unphysical results (i.e., negative concentrations). In some cases, complete separation of different species was not possible due to large numbers of SV and relatively low signal-to-noise values indicated by high *χ*^2^ error (Appendix A). 

#### 3.6.2. Selection of Suitable Datasets

Scattering profiles for each SV were further analyzed to assess their suitability for further processing, and the results were compiled in Appendix A. Since both studied proteins are composed of two parts connected via a linker, it was expected that at least monomeric forms are flexible systems, which should be confirmed by analysis of SAXS data. To select suitable data sets, we used the following criteria: reasonable values of radius of gyration (*R*_g_), *M*_w_ expected for a given fraction, e.g., monomer, dimer, and overall shape of *P*(*r*) function (Figure 6). 

*P*(*r*), also called the pair-distance distribution function, is a measure of the frequency of interatomic vector lengths within a scattering particle, providing basic information about the shape of the particle. *R*_g_ values were obtained from either Guinier analysis or from the *P*(*r*) function. For most datasets, *R*_g_ determined from *P*(*r*) was slightly higher than Guinier’s approximation, indicating conformational flexibility or the presence of oligomers. Substantial differences between these values were observed in some datasets, suggesting aggregation or denaturation. We calculate *P*(*r*) functions using a fully automated procedure based on Bayesian Indirect Fourier Transform (BIFT), which always yields one *P*(*r*) function of the highest probability. The Bayesian approach to the Indirect Fourier Transform (IFT) allows us to obtain an objective estimate of the *P*(*r*) without any human interaction, significantly reducing ‘qualified biasing’ of the statistical parameters in terms of the maximum distance within the particles (*D*_max_), and the smoothness constraint *α*. Hence, the BIFT algorithm provides a more objective estimation of *P*(*r*) compared to standard IFT algorithms [43]. Datasets indicating severe or medium aggregation were excluded from further processing (Appendix A). On the other hand, in each sample, at least one dataset displays well-behaved *P*(*r*) functions, which was the vital criterion in this assessment. For each dataset, *M*_w_ was calculated using several different methods (based on Porod volume *V*_p_, the volume of correlation *V*_c_, Bayesian inference, and comparison to known structures, See Materiał and Methods for details), and for most datasets, they converged to similar values. In our case, the lowest estimates for *M*_w_ are in the 80–90 kDa range, while the real *M*ws are significantly lower (~52 kDa). *M*_w_ for flexible or disordered systems determined from *I*(*q*) profiles tends to be overestimated due to higher apparent volume in the solution [60]. On that ground, we assumed that low-mass fractions contained mostly monomers, and the mass error was caused mainly by their dynamic behavior rather than contamination by larger molecules. It should also be emphasized that in SAXS and other scattering techniques, even a small contribution of massive particles leads to a noticeable overestimation of molecular volume, *M*_w_, and shape parameters such as *R*_g_ or persistence length, also reflected in our data [60]. 

Considering the abovementioned criteria, we selected several datasets for more detailed studies. The analysis of SV for sample 1.0 showed its higher diversity than other samples subjected to SEC-SAXS, since it contained mono- and dimeric protein and oligomers and aggregates. Dataset 1.2 contains oligomers, and 1.3 (and its truncated version 1.3T)) is mainly composed of monomers and some contributions of dimers and oligomers, making it suitable for modeling. Dataset 1.4 contains mostly trimers/tetramers and was also selected for modeling. In the case of 2.0, only a basic background correction was performed, and a set of few scattering profiles was selected based on the shape of its *P*(*r*) function, indicating the presence of only one flexible species. Fractions 3. 2 and 4.3 are composed of a mixture of oligomers and aggregates, and after EFA, one most promising elution profile from each fraction (3.2 and 4.1) was selected for modeling and ensemble analysis. Dataset 5.0 is the sample of Exo protein-containing aggregates, and an elution profile 5.1 corresponding to monomeric protein was extracted. Other datasets were discarded from further processing.

#### 3.6.3. Further Analysis of SAXS Data for Selected Datasets

There are a number of indicators of flexibility and disorder derived from I(q) profiles, including numerical parameters such as Porod exponent d and the general shape of Porod-Debye (PD), Kratky, or Kratky-Debye plots [61]. In our assessment, we firstly analyzed the shape of PD and dimensionless Kratky plots (Appendix A). Datasets of lower M_w_ (1.3, 1.3T, 1.4, 2.0, and 5.1) display partial flexibility, typical for multidomain proteins, while systems having high *M*_w_ (3.2 and 4.3) are more rigid. Shapes of PD plots also indicated that low *M*_w_ datasets contain flexible particles. A plateau in the PD plot is expected for globular systems, while it is not present for flexible or disordered molecules. The volumetric analysis allowed us to obtain d, Porod Volume (*V*_p_), and particle density ρ (Appendix A, Appendix A). Lower values of d (<3) indicate flexible or disordered protein chains, while values above 4 indicate rigid particles. Molecular shapes estimated from d correspond well to ones estimated from Kratky and PD plots. *V*_p_ values for 1.3, 1. 3T, 1.5, and 2.0 are considerably lower, suggesting they are composed mainly of monomers or dimers (in the case of 2.0), while others are oligomeric. Low values of ρ also suggest flexible systems since this parameter is determined from *V*_p_, which is overestimated for such particles. Finally, integrated total scatter intensity plots over q have a slight slope at high q values, indicating a lack of severe aggregation or interparticle interferences [39]. Analysis of scattering data in real space via the) *P*(*r*) function also provides information about the shape and behavior of scattering particles. Here, these are in agreement with the reciprocal space analysis discussed earlier (Figure 6 and Appendix A). Elongated tail for high *D*_max_ values indicates a contribution from oligomers or aggregates in the 1.3 dataset and, to a lesser degree, for 1.4, 2.0, and 5.1. All mentioned datasets possess right-skewed pair-distribution suggesting partial flexibility or the presence of two connected domains. The P(r) shape for the 3.2 dataset indicates a more rigid structure resembling a hard sphere. Dataset 4.2 possess elongated P(r) distribution skewed to very high *D*_max_ values due to significant aggregation. Moreover, it confirms that some datasets (notably 1.3 and 4.2) are contaminated by aggregates, which may hinder structural modeling. Hence, we recalculated P(r) for 1.3 datasets with additional restraint for very high (>150 Å) *D*_max_ values. As a result, we obtained truncated dataset 1.3T, in which the P(r) function is almost the same as the unrestrained one, except for a significantly shortened tail. Similar recalculation failed for 3.2, yielding unphysical P(r) functions; thus, we decided to use the unmodified function for molecular modeling.

#### 3.6.4. Electron Density Reconstruction and Ensemble Analysis

For 3D shape reconstruction, we decided to use the DENsity from Solution Scattering (DENSS) method to obtain electron density (ED) of particles directly from 1D scattering data, avoiding many assumptions limiting the accuracy of other modeling algorithms [42]. Since such reconstructions are non-uniform in terms of ED distribution, DENSS is more suitable to work with flexible or intrinsically disordered proteins than bead models. Flexible regions manifest as regions of partial occupancy where average ED is reduced. Datasets 1.3T, 1,4, 2.0, 3.2, 4.3, and 5.1 were subjected to DENNS, and in each case, except 4.3, it yielded models of satisfying quality (Appendix A, Appendix A). The overall resolution (FSC cut-off 0.5) is moderate (~40 Å for 1.3T and 5.1 or ~55 Å for 1.4 and 2.0) to low (~100 Å for 3.2), which is caused by the innate flexibility of the molecules as well as minor data contamination by scattering from high *M*_w_ particles, artificially increasing the *D*_max_ values. In the case of 4.3, the very high *χ*^2^ score indicates the reconstruction is not valid, thereby not acceptable for analysis. 

Several X-ray structures of bacterial toxin and interleukin-receptor complexes are available in the PDB database. Nonetheless, due to the internal flexibility of our chimeric proteins, their structures in solution may significantly vary from the available X-ray structures adapting a set of different conformations. In a traditional approach, high-resolution sampling of conformational space at the atomic level is achieved through molecular dynamics (MD) or Monte Carlo simulations. On the other hand, when working with lower resolution SAXS data, either a coarse-grained approach or simplified MD often dispenses an acceptable resolution for a significantly reduced computational cost. We employed several different methods of ensemble generation and compared the results. In the BILBOMD method, a fast MD simulation of interdomain linkers at a very high temperature samples conformation space and generates a set of conformers from which an *I*(*s*) curve is calculated and compared to the experimental data [62]. Later, a genetic algorithm performs a minimal ensemble search (MES) which provides a few subsets of conformers best fitting to the experimental scattering profile. Another approach to address conformational heterogeneity is to probe the space of the *φ* and *ψ* dihedral angles for previously defined flexible fragments using the Rapidly Exploring Random Trees algorithm, which is implemented in the MultiFOXS webserver [48]. When a vast set of conformers is created, a *I*(*s*) curve is calculated for each structure, and multi-state models are validated against the scoring function to select the most promising models. Another method is SREFLEX, a hybrid approach applying Normal Mode Analysis (NMA) to generate a large set of atomic models and then refine the conformation by comparing their scattering profile to the experimental data [63]. Contrary to previous methods, SREFLEX does not compute a true conformational ensemble, but a set of different conformers gives similar scattering profiles to the experimental one. 

We prepared atomic models of our proteins from several PDB structures, which were merged, rebuilt, and further optimized using local MD. These steps were necessary since improper input structures may result in computational errors or distorted output multi-stage models. All ensemble methods gave similar results, which is reflexed in the comparison of the *P*(*r*) function of computed multi-states to experimental *P*(*r*) functions for both 1.3T and 5.1 datasets (Appendix A, Appendix A). The results for the 1.3T dataset will be discussed first. BILBOMB generated several multi-state models consisting of 1 to 3 structures. For all ensembles, the *P*(*r*) shapes suggest the presence of two separate regions (two distinct maxima of the probability distribution). The *D*_max_ values for each ensemble are much lower than for the experimental *P*(*r*) curve since it does not take into account traces of oligomers presented in the data. Results obtained from SREFLEX are similar to BILBOMB, since the superposition of most relevant conformers resembles three-state models generated by the latter method. MultiFoXS generated similar multi-state models to these from BILBOMB, but *D*_max_ values are in better accordance with experimental results. IL-13 domain is located at a higher distance from DT390, which explains higher *D*_max_ values and suggests that the linker exists mainly in extended conformation providing maximal separation of the two regions. This is expected to facilitate intermolecular interaction between protein molecules leading to the formation of covalent dimers. Two-state and three-state MultiFoXS structural models were aligned to reconstruct ED from the 1.3T dataset (Figure 7A) at the density threshold necessary to recreate ED volume similar to *V*_p_ obtained from the volumetric analysis. For analysis of other ED reconstructions, we applied the same ED threshold. The superposition of the structural model and ED shows overall good structural alignment, in which DT390 fits into a bulkier fragment of ED. 

In contrast, a smaller IL-13 domain occupies an oblong and thinner ED fragment. Despite the high flexibility of the interdomain linker, the ED coverage for IL-13 is still almost complete suggesting that statistical ensembles probed a significant part of the conformational space. Ensembles computed for the 5.1 dataset, representing monomer of Exo protein, indicate reduced conformational flexibility if compared to 1.3T (Appendix A). Both BILBOMB and MultiFoXS generated only single-state models, and models from SREFLEX display a high level of resemblance to one another. All methods provided comparable results, which are in good agreement with the experiment. The only exceptions are some models from BILBOMB and MultiFoXS, whose *P*(*r*) deviates from the experimental data. Lack of either high-order molecular ensembles or conformational variety indicates reduced flexibility of the system compared to DT390-IL-13. These results also partially explain the lack of covalent dimers in the case of the Exo molecule, since the relative proximity of two domains impedes closer intermolecular interactions necessary to form covalent bonds. Without additional studies, it is difficult to explain the reduced flexibility of the Exo protein, since even the relatively short interdomain linkage provides significant conformation freedom. Structural alignment of the atomic model generated by MultiFoXS with reconstructed ED shows visible similarity indicating general agreement between these methods (Figure 7), with regions of higher electron density fitting considerably well to the main polypeptide chain (Appendix A).

Structural characterisation of protein complexes using only SAXS data usually is difficult and poses a number of challenges, especially for flexible systems [60]. One method suitable for modelling dimeric complexes is webserver FoXSDock which allows for protein-protein rigid docking restrained by a scattering profile of the complex [48]. FoXSDock generates a list of dimeric complexes sorted by a combined SAXS and statistical potential energy score. We applied this tool to obtain estimated models of DT390-IL-13 dimers using selected models from previously generated statistical ensembles. Since MultiFoXS ensembles were closest to experimental data we decided to select most prevalent conformers from this method, and following pairs of structures: *e1-1* + *e1-1*; *e2-1* + *e2-2*; *e3-2* + *e3-2* and *e4-2* + *e4-3* were subjected for modelling (see Appendix A) using *I*(*s*) profile of dataset 2.0. Since FoXSDock is unable to model covalent complexes, the possible output is only a rough approximation of the real covalent dimer and should be investigated with caution. To assess the quality of the dimers two criterions were taken into account: overall FoXSDock score and chemical plausibility giving the higher importance to the latter one. We discarded all complexes in which Cys residues were at large distances preventing them from formation disulfide bonds. Complexes possessing Cys residues in relative proximity allowing them to form covalent bonds after minor conformational adaptations such as repositioning of flexible loops or changes in their relative orientation were further evaluated on the base of their *P*(*r*) functions (Appendix A). Among these complexes, it is possible to propose two different class of dimers which vary in their presumed topology of disulfide bonds: covalent bonds between two DT390 regions (called here *endo*-dimers) or bonds between DT390 and IL-13 (dubbed as *exo*-dimers). Noticeably, for one complex composed of two *e4-3* conformers, its *P*(*r*) function is almost identical to the one computed from experiment (Figure 2C). It suggests that the real covalent dimer may possess very similar structure to the proposed endo-dimer depicted in Figure 7. The formation of disulfide bridges would require only a rotation (~30–40°) of one monomer in regard to its partner, to reduce the distance between Cys residues (Appendix A) resulting in a better fit to the ED model. Fitting of the atomic model of that dimer into a reconstructed ED strengthen this claim. The overall structural alignment is fairly high, and the polypeptide chain of more rigid fragment comprising DT390 domains fits into a region of higher density (Figure 7B). On the other hand, the fitting of the IL-13 domain is worse, which is partly caused by its flexibility. If one DT390-IL-13 was rotated relative to the second molecule, as was postulated earlier, the fit of flexible parts would be considerably improved.

Structures of endo- and exo-dimers can be considered as composite blocks for covalent oligomers, which was already proposed on the ground of SDS-PAGE and MALS results. Due to steric restrains, any oligomeric structures are likely to be composed of molecules displaying both topologies of disulfide bonds forming an endo-endo-exo trimeric structural unit, a which structure can be approximated by the partial overlay of endo- and exo-dimers. Because of the lack of high-resolution structural data, we cannot exclude the existence of high-order complexes connected mainly via DT390 domains in an endo-endo fashion. However, it would be challenging to avoid significant structural clashes between relatively bulky and rigid DT390 domains in such a case. It is unlikely that any higher-order oligomers could be formed only via DT390 domains. Interestingly, the ED model for dataset 1.4 can be reasonably fitted to the approximate structure of such a trimer composed of molecules connected via DT390 and IL-13 (Appendix A). Reconstruction of ED for a 4.3 dataset was only partially successful due to the limited resolution. It probably represents an ensemble of higher oligomers with a more rigid structure than monomers and dimers. Such compact structure requires a higher number of spatial restraints than flexible linear oligomeric chains, suggesting that higher oligomers may be cyclic or composed of partially misfolded protein chains forming globular particles via several non-specific interactions. These claims are supported by cytotoxic assays showing their limiting potency if compared to the monomer (See Paragraph 3.8 for details). On that ground, we may propose a hypothetical model of oligomerization in which a number of DT390-IL-13 molecules form caged structures connected by DT390 and IL-13 domains (Appendix A). Nonetheless, it needs to be emphasized that such a model is highly speculative and requires further experimental validation due to the deviation of the experimental P(r) function from the theoretical one computed for our approximate structural model.

### 3.7. Disulfide Mapping and Structural Analysis

Peptide and disulfide crosslink mapping was accomplished by the bottom-up tandem MS with a semi-stochastic approach and HCD fragmentation. The overall sequence coverages were: 78% for the monomer, 68% for the dimer, and 71% for the oligomers. However, the highest-intensity peptides were detected for the enzymatic toxin domain and interleukin 13 domain (Figure 8 and Appendix A). The former includes a disulfide bridge native to the toxin C187–C202. The following regions were poorly identified: the N-terminal part of the toxin translocation domain, the IL-13 peptide with two disulfide linkages, and the C-terminal amino acids. The long tryptic IL-13 peptide (residues 418–466) containing two native disulfide linkages, C421–C449 and C437–C463, is either poorly separated during LC or weakly fragmented. 

The best-scored cystine crosslink for all samples was the toxin-native one, C187–C202. Non-native linkages, for example, C187–C187, C202–C202, C187–C463, or C202–C463, were also detected but with the collective Andromeda score lower by approximately one order of magnitude (Appendix A). This shows that the native C187–C202 bond is preserved in the oligomers and might potentially mediate linking the monomeric units. The IL-13-native crosslinks, C421–C449 and C437–C463, were poorly detected in the monomer sample. Due to the low abundance of this peptide, we cannot assuredly confirm whether these cysteines might be involved in intermolecular disulfide bonding only relying on MS data. Native disulfide network is essential for both protein folding, structure, and biological function. Since the formation of disulfide bonds is reversible, it can be considered an environmentally sensitive switch regulating protein function. Some estimations show that 4–7% of all structurally defined disulfides in PDB are possibly allosteric. The exposed disulfide bond in the diphtheria region may be classified as the allosteric type, whereas two IL-13 disulfides belong to the structural type. On the other hand, reduction in the single cystine crosslink C187–C202, as was confirmed by our studies on the purified isolated DT390 protein (Appendix A), also triggers aggregation. It is highly plausible that oligomerization mainly occurs during the renaturation step, due to stochastic conformational changes resulting in cysteine mispairing. In an innate biological environment, disulfide bond formation is also regulated by other proteins and changes in redox potential, which is often difficult to emulate in vitro. 

For a better insight into the possible dimerization mechanism, we also evaluated the DT390-IL-13 propensity to aggregation and flexibility using computational methods (see Materials and Methods for details). Despite a relatively low aggregation score for the structure, a clear hydrophobic path can be spotted near the Cys residues involved in dimerization (Figure 8C and Appendix A). Moreover, many surface residues are labile, including C202 and Cys residues belonging to the IL-13 domain. A dimer composed of conformational ensembles of DT390-IL-13 calculated independently from any SAXS-based methods and positioned towards the residues involved in cross-linking also can be well fitted into ED reconstruction (Figure 8D). Hence, it may be assumed that in the first step of the dimerization, non-covalent interactions driven by hydrophobic effect orient both monomers in a way that facilitates labile C202 residue to react with C187 residue resulting in the formation of a disulfide bond. The resulting dimers still possess a hydrophobic spot onto the DT390 region and conformationally labile IL-13, allowing further oligomerization and aggregating formation.

### 3.8. Bioactivity Functional Assays—ELISA and MTS

We analyzed the receptor-binding affinities and cytotoxic potential of the dimer and the oligomers to assess the influence of oligomerization. Sigmoidal and quadratic regression analysis of the binding ELISA with the extracellular domain of IL-13RA2 human receptor provided both EC_50_ coefficients and apparent *K*_d_ (Figure 9). 

Oligomerization caused an increase in EC_50_ from 1.9 ± 0.3 nM for the monomer to 6.4 ± 0.7 nM and 6.8 ± 1.0 nM for the dimer and oligomers, respectively. Our results for the monomer are in line with the IL-13RA2 affinities for IL-13 mutants reported earlier [64,65]. However, these findings also indicate that the affinity is significantly reduced by dimerization and oligomerization. The probable explanation are steric hindrances restraining receptor binding or misfolding of the IL-13 domain in the soluble oligomers. We also tried to determine *K*_d_ derived from the quadratic model, which reflects binding thermodynamics, assuming one-to-one interaction (Equation (2)).
(2)$$K_{d}=\frac{\left[R_{f}\right]\left[L_{f}\right]}{\left[RL\right]}\;f\;\mathrm{stands}\;\mathrm{for}\;free$$

*K*_d_ values calculated using Equation (2) are significantly lower for the monomer (67.4 pM) compared to EC_50_ derived from the sigmoidal model, and *K*_d_ for both dimers and oligomers are similar to EC_50_. Nevertheless, the monomer *K*_d_ has a large prediction error, making it more difficult to interpret than EC_50_ (see [66] for technical details). 

The cytotoxic potential was assessed on two cell lines, U-251 with a high IL-13-RA2 externalization and LN229 with a moderate one (Figure 10 and Appendix A). As could be expected, the monomeric fusion cytotoxin was highly effective and selective in reducing the cell viability with indeterminate IC_50_, though it was definitely lower than 0.1 ng (1.8 pM). The cytotoxic dimer potency was lower with IC_50_, which could be approximately 0.1 ng/mL (1.8 pM). IC_50_ of the first oligomeric fraction could be predicted to be about 1 ng (18.1 pM), so one order of magnitude lower than that of the dimer. The increase in oligomeric stoichiometries was correlated with the steady decrease in the cytotoxic potential, and fractions of the oligomeric Peak 3 containing highly scattering and unstable particles had no effect on the cell viability in the applied concentration range. It is worth noting that the cytotoxic potency for all the tested fractions was explicitly dependent on targeting IL-13RA2, since the LN229 cell line was not affected. The viability assay results are in accordance with the previous studies, with IC_50_ ranging from 0.04 to 0.23 ng/mL for the U-251 cell line [67]. Noticeably, the reduction in measured affinities is not precisely correlated with the reduction in cytotoxic potency. Both dimers and oligomers had similar, relatively low IL-13RA2 affinities. On the contrary, the evaluated cytotoxic potential decreased steadily with the increasing stoichiometries. The conclusion is that the dimers and the oligomers preserve their bioactivity to some extent. Progressive oligomerization likely leads not only to suppression of IL-13 but also DT390. 

## 4. Conclusions

Our biochemical and structural investigation suggests that the oligomerization process of the DT390-IL-13 cytotoxin is spontaneously initiated during the refolding of the denatured protein. The resulting dimers and oligomers with lower *M*_w_ are soluble, spheroidal, and flexible particles with defined shape and size parameters. They are stabilized primarily by the native- or non-native-type disulfide linkage(s). This fact and the identified polymerization pattern for the smaller oligomers suggest some specificity of the process. Interestingly, the oligomerization mechanism involves intermolecular covalent disulfide bridging and probably weak non-covalent intermolecular surface interactions. The question remains on their comparative role and significance on both initialization and subsequent stabilization of the dimers and oligomers. Reduced receptor-affinity and cytotoxic potential indicate that dimerization and further oligomerization may impair IL-13 ligand binding with IL-13RA2 receptor or physiological functions of the toxin molecule, including efficient toxin translocation into the cell cytosol, efficient reduction of a disulfide bond in the cytoplasm, and ADP-ribosylation of translation elongation factor. However, generally, the oligomerization diminishes the efficiency of manufacturing and quality of the cytotoxic protein preparation and should be reduced. Our results indicate that blocking intermolecular interactions and promoting intramolecular ones during the refolding process would potentially shift the equilibrium from the dimeric/oligomeric toward monomeric cytotoxin. This emphasizes the need for multidimensional optimization of many parameters, including ionic strength, pH, osmolality, and amino-acid substitutions. Whether refolding conditions could be optimized to further shift the equilibrium from intermolecular toward intramolecular association is an open question. Such an endeavor would require a significant body of work since, in most situations, a trial-and-error approach is the only viable option. The optimization of the fusion proteins production process and more efficient monomer fraction isolation is currently under scrutiny in our laboratory.

## Figures and Tables

**Figure 1 biomolecules-12-01111-f001:**
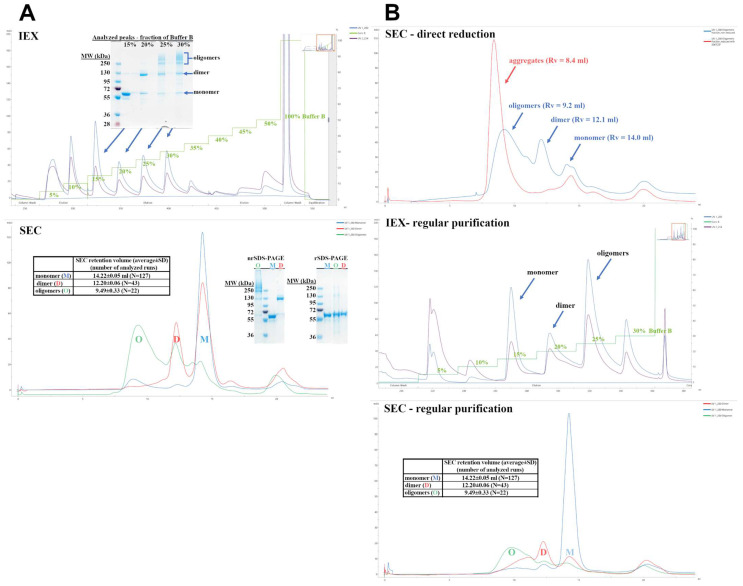
IEX and SEC chromatography of the cytotoxin. (**A**). IEX chromatogram with the recorded absorbance (254 nm, purple; 280 nm, blue) and conductivity (green) traces presents the initial separation of the protein dialysate into monomer, dimer, and oligomers. The peak fractions were resolved by non-reducing SDS-PAGE, revealing electrophoretic mobility of the monomer and dimer protein bands and a ladder of bands for the oligomer fractions. SEC chromatograms of the separated IEX peak fractions with the overlaid 280 nm traces are presented for the monomer (blue), dimer (red), and oligomers (green). The table inlet contains average elution volumes for each molecular state. The apex fractions were resolved by non-reducing and reducing SDS-PAGE. (**B**). The cytotoxin recovery from the oligomeric fractions by two tested approaches, direct reduction and regular purification protocol with IEX and SEC chromatography. Analytical SEC of the reduction substrates (blue) and products (red)—shifting the elution volume (*V*_SEC_) for the monomer, dimer, and oligomers toward column void volume (about 8.0 mL) indicates aggregation. IEX chromatogram of the solubilized and refolded oligomers, separated into monomers, dimers, and oligomers. SEC chromatograms of the separated IEX peak fractions with the overlaid 280 nm traces are presented for the monomer (blue), dimer (red), and oligomers (green). The table inlet contains average elution volumes for each molecular state. X axes indicate elution volume (mL).

**Figure 2 biomolecules-12-01111-f002:**
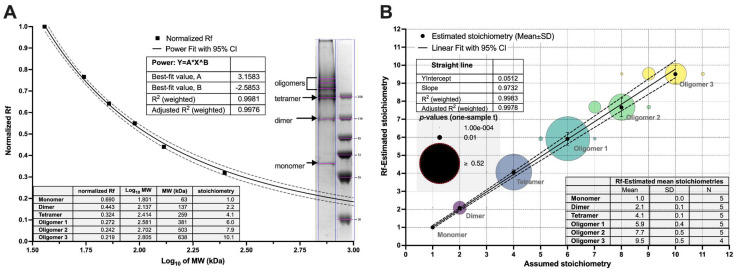
Determination of oligomers stoichiometry in non-reducing SDS-PAGE gels by analyzing relative front parameter (*R*_f_). (**A**). Exemplary non-linear regression of the standard molecular weights plotted against their normalized relative fronts. The power model parameters were used to derive an apparent molecular weight of the cytotoxin bands, and their stoichiometries averaged for several gel paths. (**B**). The linear correlation between *R*_f_-estimated and assumed stoichiometries for the dimer and oligomers. The *R*_f_ value for each band was also compared to the theoretical stoichiometries by the one-sample *t*-test, and the resulting *p*-values are displayed on the bubble plot. Each band was compared to three or four most probable values. Dotted lines represent the 95% confidence intervals (CI) for the regression models.

**Figure 3 biomolecules-12-01111-f003:**
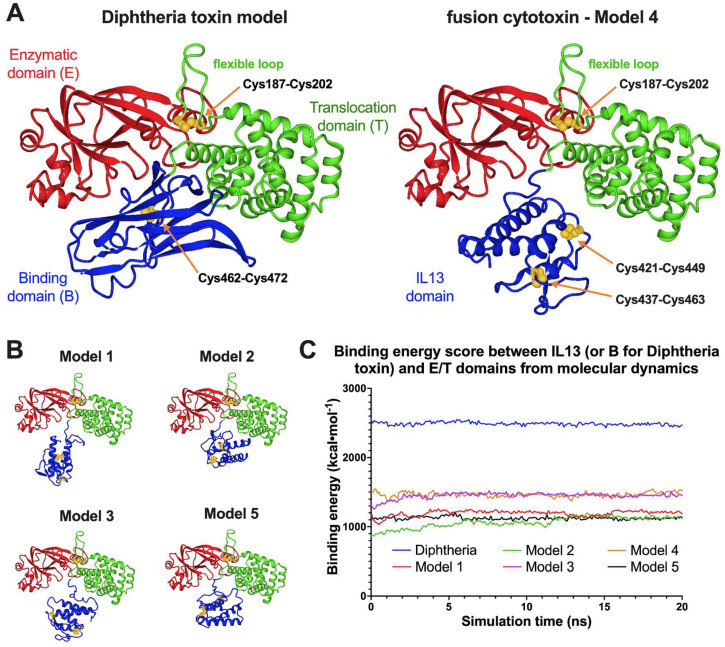
Molecular dynamics simulations for the diphtheria toxin and the fusion cytotoxin monomer. Presented are the ribbon representations of the last snapshots of the 20 ns MD. (**A**). Diphtheria toxin model compared to the cytotoxin Model 4, the one with the highest average DT390 and IL-13 interdomain binding energy score. (**B**). Structures of four other constructed models. (**C**). Trajectories of DT and DT390-IL-13 models as “binding energy” parameter in the function of time.

**Figure 4 biomolecules-12-01111-f004:**
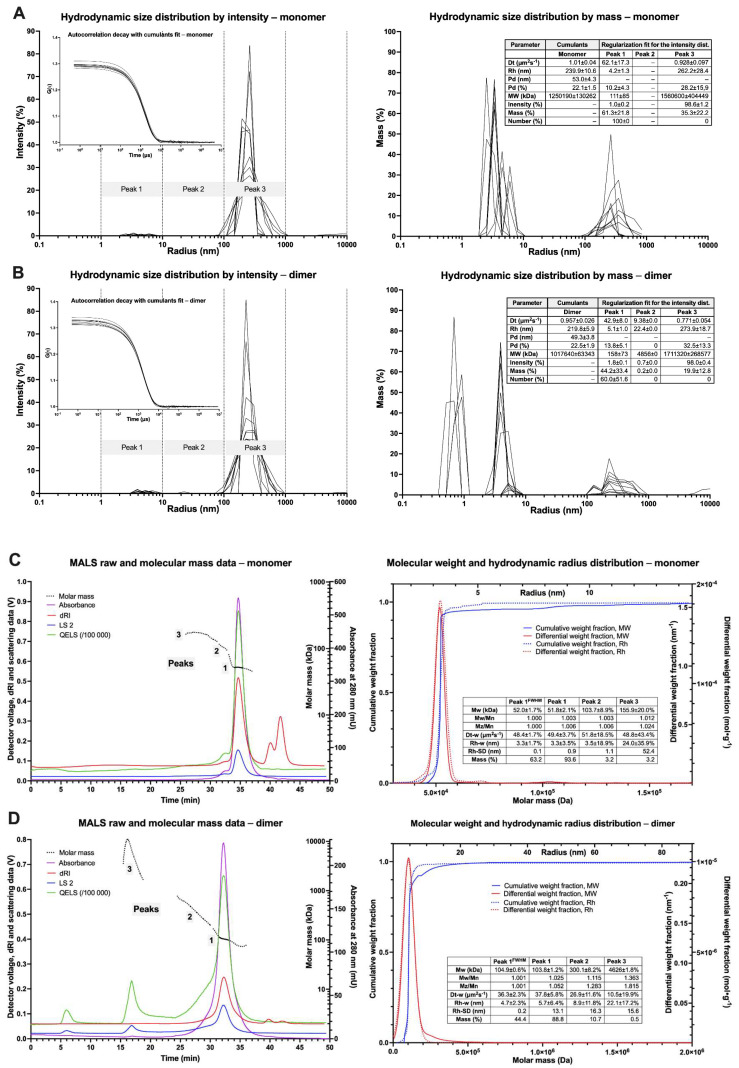
Results of DLS and MALS measurements for monomeric and dimeric fractions of DT390-IL-13. (**A**) DLS of monomeric fractions. (**B**) DLS of dimeric fraction. (**C**) MALS of monomeric fractions. (**D**) MALS of dimeric fraction.

**Figure 5 biomolecules-12-01111-f005:**
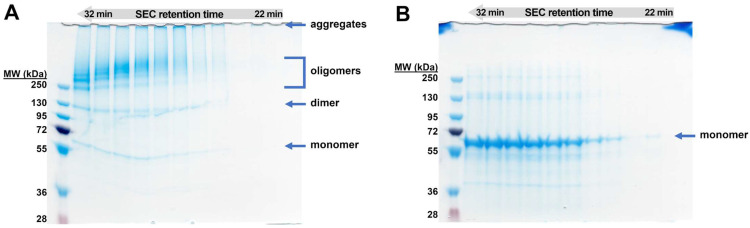
SDS-PAGE analysis in non-reducing (**A**) and reducing (**B**) conditions of the consecutive SEC-MALS oligomer fractions with the elution time range of 22–32 min. An amount of 10 µg was loaded into each well. The average mass distribution over the analyzed fractions in the non-reducing conditions is inversely proportional to the SEC-MALS elution time, as expected. The higher the molecular weights, the more unstable the cytotoxin. The oligomeric fractions additionally contain monomer, dimer, and aggregates. The efficient oligomers monomerization was observed in the reducing conditions.

**Figure 6 biomolecules-12-01111-f006:**
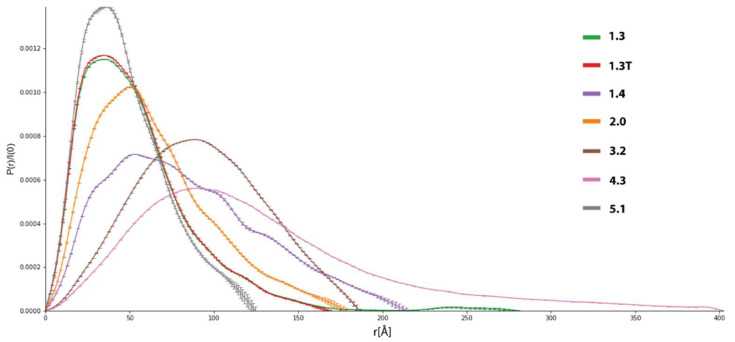
*P*(*r*) functions for selected datasets extracted from initial SEC-SAXS data. All these datasets were selected for further analysis.

**Figure 7 biomolecules-12-01111-f007:**
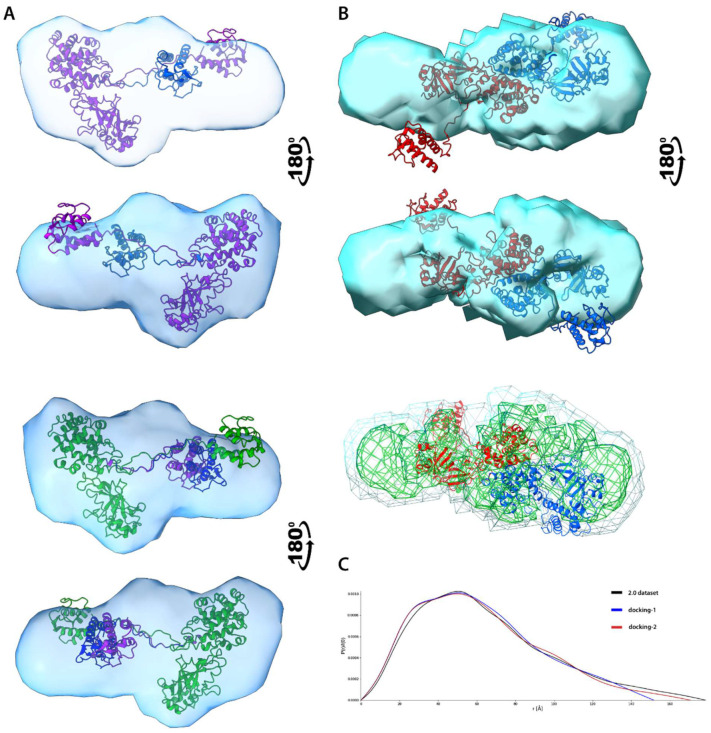
SAXS structural modelling of DT390-IL-13. (**A**). Alignment of atomic models representing 2-state (top) and 3-state (bottom) models onto ED reconstruction. Each conformer is depicted using a different color. (**B**). Alignment of atomic model of best-scoring dimer onto ED reconstruction, each monomer is depicted using a different color (top). Layers of different electron density showing coverage of polypeptide chain (bottom). (**C**). *P*(*r*) functions for both original and 2.0 dataset (representing a dimer)and two selected dimers created by the FoXSDock.

**Figure 8 biomolecules-12-01111-f008:**
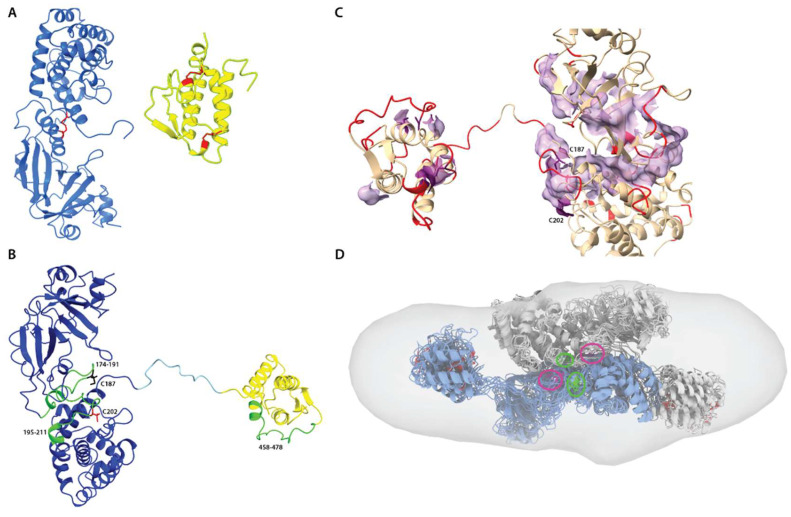
Disulfide mapping and protein-protein recognition. (**A**). Native disulfide bridges (red) in DT390 domains (blue) and IL-13 (yellow). (**B**). Cysteine-containing peptides identified by MS (green). Reduced C187 and C202 residues are colored black and red, respectively. (**C**). Fragment of DT390 with flexible residues colored red and surface of the major hydrophobic path. Residues C187 and C202 are highlighted in purple and belong to the hydrophobic path. (**D**). Fitting conformational assemblies of DT390-IL-13 dimer to ED model. Each “monomer” is a structural alignment of 10 most probable conformers, and both molecules are manually docked to achieve proximity between C187 and C202 residues highlighted in green and purple, respectively.

**Figure 9 biomolecules-12-01111-f009:**
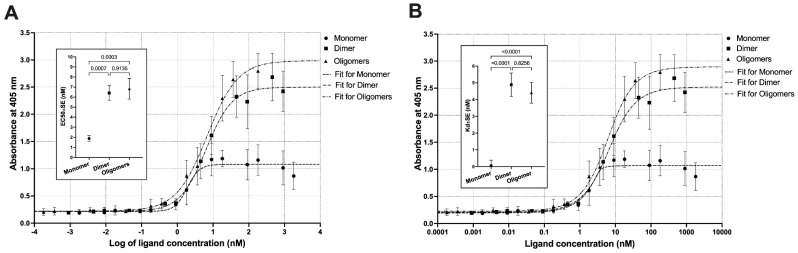
Non-linear regression of the ELISA experimental data for the titration of IL13RA2 receptor with the cytotoxin monomer, dimer, and oligomers. Two models were used: (**A**) sigmoidal, to derive EC_50_, and (**B**) quadratic, to derive apparent *K*_d_ values. The inlet graphs present the fit parameters, EC_50_ or *K*_d_, and *p*-values for the pairwise comparisons by one-way ANOVA.

**Figure 10 biomolecules-12-01111-f010:**
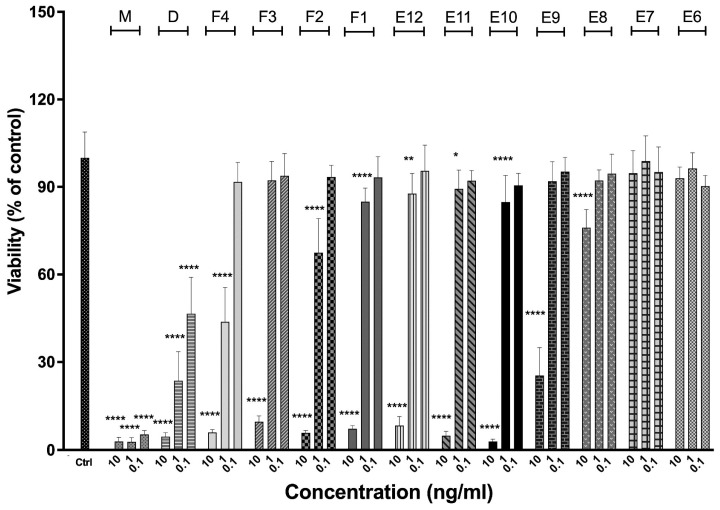
The viability of U-251 glioblastoma cell line in presence of the cytotoxin monomer (M), dimer (D), and oligomers (F4-E6), measured by the MTS assay, in three concentrations, 0.1, 1.0, and 10.0 ng (which correspond to 1.8, 18.1, and 181.3 pM concentration, respectively). Significant differences are indicated as: for * *p* < 0.05, ** *p* < 0.01 and **** for *p* < 0.001.

## Data Availability

The data presented in this study are available on request from the corresponding author.

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
