# Peer review of "Structure–Activity Relationship of the Dimeric and Oligomeric Forms of a Cytotoxic Biotherapeutic Based on Diphtheria Toxin"

_biomolecules, 2022, doi:10.3390/biom12081111_

Round 1

Reviewer 1 Report

The authors present a very detailed biophysical study of the aggregation of a reengineered toxin. The experiments have been selected well to achieve the authors objectives - perhaps some parts of the manuscript have excessive amounts of data (for example the SAXS study, but I must admit that is the section I am least well qualified to comment on). The protein expression and chromatographic purification/analysis is described very well. The molecular dynamics is appropriate, but see comment below that needs correction/clarification. The SAXS analysis I cannot comment on, the binding experiments appear to have been conducted correctly but I have a concern about the validity of the Kd values reported. The biological analysis shows what one might anticipate - that the monomeric form of the protein has best activity. I would encourage the authors to try to be a bit more concise, but I recommend publication after addressing a few minor points:

1. line 114 - would "exploration" be a better word here than "explanation" - that did not make so much sense to me.

2. Figure 2 - please provide clarification of the meaning of the dotted lines in the figure caption. I presume they are some confidence level on the fitted line, but this could be stated more clearly.

3. line 539 - would "extended" eb a better word here than "prolonged"?

4. line 539 - Figure 3 does not indicate that model 4 is the most energetically stable - quite the contrary it appears to be the highest energy of the reengineered proteins. model 2 looks lower energy

5. line 971 - the 67 pM Kd figure does not make sense when looking at the graph in figure 9b. That would correspond to a point on the lower asymptote of the fitted line rather than in the transition region. As the curve for the monomer turns the corner very abruptly between 1 and 10  nM, I think this shape of curve indicates that the receptor concentration is higher than Kd for the interaction and so the assumption in the binding model that free ligand concentration is approximately equal to total ligand concentration will no longer hold and that might be why the fitting is failing to give a reasonable value for Kd. I suggest removing this fitting method and just presenting the EC50 values which look more reasonable.

Author Response

Reviewer 1

The authors present a very detailed biophysical study of the aggregation of a reengineered toxin. The experiments have been selected well to achieve the authors objectives - perhaps some parts of the manuscript have excessive amounts of data (for example the SAXS study, but I must admit that is the section I am least well qualified to comment on). The protein expression and chromatographic purification/analysis is described very well. The molecular dynamics is appropriate, but see comment below that needs correction/clarification. The SAXS analysis I cannot comment on, the binding experiments appear to have been conducted correctly but I have a concern about the validity of the Kd values reported. The biological analysis shows what one might anticipate - that the monomeric form of the protein has best activity. I would encourage the authors to try to be a bit more concise, but I recommend publication after addressing a few minor points:

  • Thank you for the positive comment and appreciation of our efforts. We tried to introduce all possible for us experimental tools to verify our hypothesis. Please, find below our responses to Your comments.
  1. line 114 - would "exploration" be a better word here than "explanation" - that did not make so much sense to me.

- Thank You for noting. It was amended accordingly. 

  1. Figure 2 - please provide clarification of the meaning of the dotted lines in the figure caption. I presume they are some confidence level on the fitted line, but this could be stated more clearly.

- Exactly, the dotted lines are 95-% CI of the regression models. Explanation was added. 

  1. line 539 - would "extended" eb a better word here than "prolonged"?

- Thank You for noting. It was amended accordingly. 

  1. line 539 - Figure 3 does not indicate that model 4 is the most energetically stable - quite the contrary it appears to be the highest energy of the reengineered proteins. model 2 looks lower energy

- Let us, please, explain. To quantify the possible dynamic interaction between the domains IL-13 and DT390 we used the BindEnergyObj command implemented in YASARA. This software takes the convention that “The more positive the binding energy, the more favorable the interaction in the context of the chosen force field”. Binding energy, usually positive, is defined here as the energy required to disassemble a whole into separate parts. We added suitable explanation to the Section 2.5. 

  1. line 971 - the 67 pM Kd figure does not make sense when looking at the graph in figure 9b. That would correspond to a point on the lower asymptote of the fitted line rather than in the transition region. As the curve for the monomer turns the corner very abruptly between 1 and 10 nM, I think this shape of curve indicates that the receptor concentration is higher than Kd for the interaction and so the assumption in the binding model that free ligand concentration is approximately equal to total ligand concentration will no longer hold and that might be why the fitting is failing to give a reasonable value for Kd. I suggest removing this fitting method and just presenting the EC50 values which look more reasonable.

- Thank You for paying attention to the issue. This is a general difficulty of reliably measuring high affinities in a typical biochemistry lab setting, applying equilibrium (not kinetic) binding measurements with ELISA. Simple dose-response/sigmoidal model when not conjugated with the strict concentration regime [Rt]<<<Kd is prone to artifacts, when comparing solely the derived EC50 parameters. This is the reason our EC50 for monomer (about 1.8 nM) was just approximately half the total concentration of the immobilized receptor protein (about 3.5 nM) – it just titrated the receptor. This can be partially overcome by applying the quadratic model, as an advanced analysis method, which explicitly accounts for the binding thermodynamic term [L][R]/[RL] and is not as much sensitive to concentration regimes as the simple dose-response model. Thus, the applied quadratic model is, in fact, necessary to have any reliable estimation of the real affinities. However, when [Rt]>>>Kd, the lower the Kd values the more difficult to discriminate they are and more sensitive to statistical noise. Therefore, the lower confidence interval for the monomer Kd was undefined. However still, our results clearly indicate that Kd for monomer is much lower than the affinities for dimer and oligomers and compares well with the reported values. That’s why we consider this analysis a valuable estimation of the real affinities and would be very happy to retain it in the manuscript. Unfortunately, we do not currently have any access to an SPR instrument, which would allow to calculate affinities more reliably based on the kinetic mode (not equilibrium). 

Reviewer 2 Report

This is an interesting study as it aims at characterizing oligomerization issues of cytokine-toxin production, which cytokine-toxin may have therapeutic potential. These issues are a real concern for the development of such biotherapeutics. In this article, the authors propose an in depth characterization of the oligomers, combining chromatography, SDS-PAGE, MS-MS analysis and SAXS. Although the results do not allow to fully understand the illegitimate disulfide bond formation, it truly helps to  get insight into the heterogeneity of the misfolded protein oligomers.   Minor comments:  - The use of the term "domain" is confusing. It seems the authors refer to the diphtheria toxin part of the protein as the diphtheria toxin domain while there are indeed two domains involved: the catalytic and the translocation domains.  - The convention for biotoxins derived from diphtheria toxin is to call them DT(C-ter aa number)-ILX, here DT390-IL-13. The use of the abbreviation DT for the hybrid toxin in the results and discussion section, which is conventionally used for the native diphtheria toxin is rather confusing. Especially, DT is used for the native toxin in the introduction and for the immunotoxin in the results. Please correct. - There are misspelling of some toxins, especially cintredekin besudotox (not cintradekin). Please correct.  - It would be nice to have the concentration of the toxin monomer or monomer equivalent given in molarity  (not only in ng/mL) in the bioactivity study.

Author Response

Reviewer 2

This is an interesting study as it aims at characterizing oligomerization issues of cytokine-toxin production, which cytokine-toxin may have therapeutic potential. These issues are a real concern for the development of such biotherapeutics. In this article, the authors propose an in-depth characterization of the oligomers, combining chromatography, SDS-PAGE, MS-MS analysis and SAXS. Although the results do not allow to fully understand the illegitimate disulfide bond formation, it truly helps to get insight into the heterogeneity of the misfolded protein oligomers.  

  • Thank you for the positive comment and appreciation of our efforts. We tried to introduce all possible for us experimental tools to verify our hypothesis. Please, find below our responses to Your comments.

Minor comments: 

- The use of the term "domain" is confusing. It seems the authors refer to the diphtheria toxin part of the protein as the diphtheria toxin domain while there are indeed two domains involved: the catalytic and the translocation domains. 

- Indeed, the cytotoxin under study is composed of two N-terminal diphtheria toxin domains and interleukin 13. We inappropriately used the term “domain” to describe the DT390 region of the cytotoxin. The use of the term was revised across the manuscript. 

- The convention for biotoxins derived from diphtheria toxin is to call them DT(C-ter aa number)-ILX, here DT390-IL-13. The use of the abbreviation DT for the hybrid toxin in the results and discussion section, which is conventionally used for the native diphtheria toxin is rather confusing. Especially, DT is used for the native toxin in the introduction and for the immunotoxin in the results. Please correct.

- Thank You for the suggestions. The use of abbreviation DT in the manuscript was limited to the full-length native diphtheria toxin. 

- There are misspelling of some toxins, especially cintredekin besudotox (not cintradekin).

- Thank You for the suggestions. The manuscript was corrected accordingly. 

Please correct.  - It would be nice to have the concentration of the toxin monomer or monomer equivalent given in molarity (not only in ng/mL) in the bioactivity study.

- Thank You for the suggestions. The manuscript was corrected accordingly. The molar concentrations of the cytotoxin in the bioactivity study were calculated parallelly to the already given mass-to-volume concentrations: 0.1 ng/ml = 1.81 pM, 1 ng/ml  = 18.1 pM, 10 ng/ml = 181.3 pM. This may be handful in interpreting the results. 

Reviewer 3 Report

The MS was well conducted. The monomeric and oligomeric state of DT was analyzed by IEX, SEC, light scattering, MD and SAX. Three minor points could be addressed:

1. SAX analysis section contains too much technical information that could be improved by writing in simpler terms.

2. Binding affinity and activity were analyzed with M, D and O. Those results showed that D and O lost affinity and activity. Authors attributed this effect to the state aggregated by disulfide bridges, however D and O must be analyzed under reducing conditions to verify this hypothesis.

3. Authors claim that the affinity was not be determined from the data in Figure 9B. In my point of view, the scatchard equation could be used to solve the problem.

Author Response

The MS was well conducted.The monomeric and oligomeric state of DT was analyzed by IEX, SEC, light scattering, MD and SAX.Three minor points could be addressed:

- Please, find below our responses to Your comments.

  1. SAX analysis section contains too much technical information that could be improved by writing in simpler terms.

- According to the Reviewer suggestion, SAX analysis section has been corrected and detailed technical information partially removed to the Supplementary Materials.

  1. Binding affinity and activity were analyzed with M, D and O. Those results showed that D and O lost affinity and activity.Authors attributed this effect to the state aggregated by disulfide bridges, however D and O must be analyzed under reducing conditions to verify this hypothesis.

- Unfortunately, the protein immediately precipitates in reducing conditions, what essentially makes such an analysis infeasible. This demonstrates the structural and stabilizing role of our disulfide bridges. 

  1. Authors claim that the affinity was not be determined from the data in Figure 9B. In my point of view, the scatchard equation could be used to solve the problem.

- Thank You for the suggestion. However, it is generally advised to avoid any data linearization, instead, the original data for non-linear regression should be modelled. Nonlinear regression produces the most accurate results. Please, refer for example to: https://www.graphpad.com/guides/prism/latest/curve-fitting/avoidscatchard_lineweaver_burkeandsimilartransforms.htm 

The problem with the uncertainty of the Kd parameter for the monomer is linked to the general difficulty of reliably measuring high affinities in a typical biochemistry lab setting, applying equilibrium (not kinetic) binding measurements with ELISA, even when conjugated to the more advanced quadratic model, which explicitly accounts for the binding thermodynamic term [L][R]/[RL] and is not as much sensitive to concentration regimes as the simple dose-response model. However, in the concentration regime [Rt]>>>Kd, the lower the Kd values the more difficult to discriminate they are and more sensitive to statistical noise. Therefore, the lower confidence interval for the monomer Kd was undefined. However still, our results clearly indicate that Kd for monomer is much lower than the affinities for dimer and oligomers and compares well with the reported values. Unfortunately, we do not currently have any access to an SPR instrument, which would allow to calculate affinities more reliably based on the kinetic mode (not equilibrium). 

Reviewer 4 Report

The authors have analyzed different DT cytotoxin aggregation derived from the in vitro refolding after its purification.

The analysis of the different states of aggregation versus the protein in its monomeric conformation is interesting, but the results shown fail to demonstrate anything of relevant interest.

The MTS experiment shows that the monomers cause the toxicity and the dimers trigger toxicity like others protein aggregates, not because this toxin in question is aggregated. Authors do not show anything to indicate otherwise. ELISA show that monomers obviously bind better to the IL13RA2 receptor than dimers or oligomers.

I miss loading controls in the chromatographies (both in affinity and molecular exclusion), there is a lack of internal toxicity control in the MTS assay and a binding control in the ELISA.

The quality of the data presented in the manuscritp is generally very low. Also, in figure 1, all of the x-axes do not show what data is represented (elution volumen).

Author Response

Reviewer 4

The authors have analyzed different DT cytotoxin aggregation derived from the in vitro refolding after its purification. The analysis of the different states of aggregation versus the protein in its monomeric conformation is interesting, but the results shown fail to demonstrate anything of relevant interest.

  • We are very sorry that the Reviewer did not appreciate the presented manuscript. The observations presented are exploratory and relate to the problem of protein aggregation identified during attempts to synthesize a biological drug. The results presented in the manuscript were generated with the purified and separated protein using the in-depth described, standard or nearly standard biochemical and biophysical methods. The data were subjected to the thorough and comprehensive analysis. This study aims at characterizing oligomerization issues of cytokine-toxin production, which cytokine-toxin may have therapeutic potential. These issues are a real concern for the development of such biotherapeutics. We proposed an in-depth characterization of the oligomers, combining chromatography, SDS-PAGE, MS-MS analysis, and SAXS. We are aware that the results do not allow us to fully understand the illegitimate disulfide bond formation. Still, it truly helps to get insight into the heterogeneity of the misfolded protein oligomers. The availability of works describing the problem of protein aggregation is wide. At the same time, it is challenging to find papers that would indicate their causes and possible optimization to limit them. However, agreed that continuing the presented research on the cytotoxin with additional and more advanced methods might expand the so-far understanding. Nonetheless, our studies may be seen as the ones reviving this line of the cytotoxin biotherapeutics characterizing.

The MTS experiment shows that the monomers cause the toxicity and the dimers trigger toxicity like others protein aggregates, not because this toxin in question is aggregated. Authors do not show anything to indicate otherwise. ELISA show that monomers obviously bind better to the IL13RA2 receptor than dimers or oligomers.

  • MTS assay clearly shows that the most cytotoxic action exerts a monomeric fraction of IL-13-DT390 protein, whereas dimeric and oligomeric forms are less active. Furthermore, the reduction of cell viability was negatively correlated to aggregates' molecular mass. Our goal was to illustrate that protein aggregation is a negative phenomenon crucial to the biological activity of a potential drug (IL-13-DT390). ELISA assay supports our conclusion that efficient receptor bounding is disturbed when aggregates are formed. That is why we did not perform more depth-in studies to prove what is already visible from these two assays.

I miss loading controls in the chromatographies (both in affinity and molecular exclusion), there is a lack of internal toxicity control in the MTS assay and a binding control in the ELISA.

  • The cytotoxicity (MTS) and binding (ELISA) controls were performed in the course of the study, however, were not directly presented in the manuscript. The cytotoxicity control was cycloheximide, with IC50 20 uM. The chemical reduced the viability of the cell lines as efficiently as the cytotoxin monomer. The binding control was the commercially available human IL-13, which in another experiment, bound the receptor as efficiently as the cytotoxin monomer, comparing their EC50 values. The aim of the presented MTS and ELISA results was to compare the molecular form of the cytotoxin monomer in the same experiment. Therefore, we assumed that there was no point in expanding the data.

The quality of the data presented in the manuscript is generally very low. Also, in figure 1, all of the x-axes do not show what data is represented (elution volumen).

  • The figures presenting chromatograms are automatically generated with the UNICORN software (Cytiva). The current labels, “Elution – ml”, are obviously not as clearly marked as they should be. We are sorry for the inconvenience.The additional explanation has been added in the Figure 1 legend.

Round 2

Reviewer 4 Report

I agree with the authors that the problem of protein aggregation in industry needs to be addressed. At the same time, many works can be found in the literature in which authors have had to deal with protein aggregation during purification processes. At the level of proteins with biotechnological interest, it is true that it is a challenge to find the reasons why proteins aggregate and the possible optimization to limit them. In this sense, the authors have carried out a good approach. However, part of the work of the manuscript is based on the study of the toxicity of these aggregates and the important thing is to know how to reduce them and how to enrich the monomeric fraction because the monomeric toxin is the fraction that has the ability to interact with its IL-13RA2 receptor.

Author Response

Reviewer 4

I agree with the authors that the problem of protein aggregation in industry needs to be addressed. At the same time, many works can be found in the literature in which authors have had to deal with protein aggregation during purification processes. At the level of proteins with biotechnological interest, it is true that it is a challenge to find the reasons why proteins aggregate and the possible optimization to limit them. In this sense, the authors have carried out a good approach. However, part of the work of the manuscript is based on the study of the toxicity of these aggregates and the important thing is to know how to reduce them and how to enrich the monomeric fraction because the monomeric toxin is the fraction that has the ability to interact with its IL-13RA2 receptor.

  • We fully agree with the Reviewer's opinion that the most crucial part of the manuscript and our studies is the production of cytotoxic IL-13RA2-targeting fusion protein that is able to induce GBM cell death efficiently. The manuscript describes the characteristic of the aggregates and their estimated characterization but does not provide a clear solution for protocol optimization for better monomer fraction isolation. This part of research is still under scrutiny in our laboratory and is a real challenge for us. As mentioned in the Conclusions section, "Such an endeavor would require a significant body of work since, in most situations, a trial-and-error approach is the only viable option." However, we have some ideas for protocol modification to achieve a bigger fraction of the fusions protein's monomer. To clarify this, we added the following statement: "Our results indicate that blocking intermolecular interactions and promoting intramolecular ones during the refolding process would potentially shift the equilibrium from the dimeric/oligomeric toward monomeric cytotoxin species. This emphasizes the need for multidimensional optimization of many parameters, including ionic strength, pH, osmolality, and amino-acid substitutions". I am sure that in the case of highly effective enrichment of monomeric fraction in our modified protocol, we will publish our observations in a dedicated article.